# Seasonal variation of atmospheric pollutants transport in central Chile: dynamics and consequences

Rémy Lapere[1], Laurent Menut[1], Sylvain Mailler[1,2], and Nicolás Huneeus[3]

[1]Laboratoire de Météorologie Dynamique, IPSL, École Polytechnique, Institut Polytechnique de Paris, ENS, Université PSL, Sorbonne Université, CNRS, Palaiseau, France
[2]École des Ponts, Université Paris-Est, 77455 Champs-sur-Marne, France
[3]Department of Geophysics, Universidad de Chile, Santiago, Chile

**Correspondence:** Rémy Lapere (remy.lapere@lmd.ipsl.fr)

**Abstract.** Central Chile faces atmospheric pollution issues all year long, in relation with elevated concentrations of fine particulate matter during the cold months and tropospheric ozone during the warm season. In addition to public health issues, environmental problems regarding vegetation growth and water supply, as well as meteorological feedback are at stake. Sharp spatial gradients in regional emissions along with a complex geographical situation make for variable and heterogeneous dynamics in the localization and long-range transport of pollutants, with seasonal differences. Based on chemistry-transport modeling with WRF-CHIMERE, this work studies for one winter period and one summer period: (i) the contribution of emissions from Santiago city to air pollution in central Chile, (ii) the reciprocal contribution of regional pollutants transported into the Santiago basin. The underlying 3-dimensional advection patterns are investigated. We find that on average for the winter period $5\,\mu\mathrm{g\,m^{-3}}$ to $10\,\mu\mathrm{g\,m^{-3}}$ of fine particulate matter in Santiago come from regional transport, corresponding to 13% to 15% of average concentrations. In turn, emissions from Santiago contribute to 5% to 10% of fine particulate matter pollution as far as $500\,\mathrm{km}$ to the north and $500\,\mathrm{km}$ to the south. Wintertime transport occurs mostly close to the surface. In summertime, exported precursors from Santiago, in combination with mountain-valley circulation dynamics, are found to account for most of ozone formation in the adjacent Andes cordillera and to create a persistent plume of ozone of more than $50\,\mathrm{ppb}$, extending along $80\,\mathrm{km}$ horizontally and $1.5\,\mathrm{km}$ vertically, and located several hundred meters above ground, slightly north of Santiago. This work constitutes the first description of the mechanism underlying the latter phenomenon. Emissions of precursors from the capital city also affect daily maxima of surface ozone hundreds of kilometers away. In parallel, cutting emissions of precursors in the Santiago basin results in an increase of surface ozone mixing ratios in its western area.

## 1 Introduction

Most urban areas in central Chile (or *Zona central*, extending between 32°S and 37°S), including the capital city Santiago (33.5°S, 70.7°W, $500\,\mathrm{m}$ a.s.l.), deal with harmful atmospheric concentrations of fine particulate matter ($PM_{2.5}$) in wintertime (Saide et al., 2016; Barraza et al., 2017; Toro et al., 2019; Lapere et al., 2020), and tropospheric ozone ($O_3$) in summertime (Gramsch et al., 2006; Seguel et al., 2013, 2020). Strong anthropogenic emissions of primary pollutants and precursors, combined with the poor ventilation conditions induced by the topography and synoptic-scale circulation, are the core reasons

for such air quality issues (Rutllant and Garreaud, 1995). The main sources of atmospheric pollutants in Santiago are road traffic and industrial activities, with additional contributions in wintertime from wood burning for residential heating (Barraza et al., 2017; Mazzeo et al., 2018). Such characteristics apply to most urban areas in central Chile, including coastal zones (Sanhueza et al., 2012; Toro et al., 2014; Marín et al., 2017).

The central zone of Chile investigated in this study (referred to as 'central Chile' in the continuation) comprises the six administrative regions of Coquimbo, Valparaíso, Metropolitana de Santiago, O'Higgins, Maule and Ñuble. It is home to more than 12 million people (INE, 2018), who are chronically exposed to $PM_{2.5}$ pollution, leading to respiratory and cardiovascular issues (Ilabaca et al., 1999; Soza et al., 2019). Chronic and acute exposures to $PM_{2.5}$ pollution also induce a significant economic burden (MMA, 2012; OECD, 2016). A side effect is the deposition of light-absorbing particulate matter, such as black carbon (BC), on the adjacent Andean snowpack contributing to the observed accelerated melting of glaciers (Rowe et al., 2019; Lapere et al., 2021b). Aerosols also affect the radiative balance of the atmosphere and influence cloud formation by acting as cloud condensation nuclei (e.g. Chung and Seinfeld, 2005; Koch and Genio, 2010).

Tropospheric $O_3$ is a secondary pollutant formed by the photochemical oxidation of volatile organic compounds (VOC) in the presence of nitrogen oxides ($NO_x$). The essential role of photolysis in its production explains that harmful levels are mostly observed in summertime (e.g. Walcek and Yuan, 1995; Seinfeld and Pandis, 2006). $O_3$ is noxious for human health, causing respiratory disorders such as asthma (Lippmann, 1991). Furthermore, its deposition on plant leaves affects their photosynthesis and evaporation ability, hence damaging crop yields (Hill and Littlefield, 1969). Tropospheric $O_3$ is also a powerful greenhouse gas (GHG) as well as a photochemical oxidant hence playing a key role in the atmosphere (Ehhalt et al., 2001).

Although anthropogenic urban pollution is mostly a phenomenon affected by local sources and meteorology, interactions with remote emissions and air masses also occur. Depending on the wind system and stability of the troposphere, pollutants and precursors can be transported far from the emission site and reach distant locations. Urbanized areas in Europe (Vardoulakis and Kassomenos, 2008) and South America (Resquin et al., 2018) usually feature a marked contribution of long-range transport to measured concentrations of particulate matter within urban basins. For Santiago, an example can be found in the wildfires occurring frequently in summertime in central Chile, explaining sporadic peaks of particulate matter and ozone in the city although the sources are found more in the South (Rubio et al., 2015; de la Barrera et al., 2018; Lapere et al., 2021a). In this case, pollutants are not directly of anthropogenic origin, which is out of the scope of the present work and therefore ignored throughout this study. Although specific studies revealed the importance of remote sources such as copper smelters (e.g. Gallardo et al., 2002; Hedberg et al., 2005; Moreno et al., 2010) and marine aerosols (e.g. Kavouras et al., 2001; Jorquera and Barraza, 2012; Barraza et al., 2017) in urban pollution in Chile, generally speaking the processes and patterns underlying pollutants transport are not well known, nor is the amount of advected contaminants.

Chile is a narrow band of land bordered by the Pacific ocean on the western side and the Andes cordillera on the eastern side. Air motions are thus influenced by sea-land atmospheric interactions and mountain-valley circulation, in addition to more synoptic patterns. The intensity of these atmospheric regimes, which are partly governed by radiative processes, are modulated seasonally. So do emissions of primary pollutants and photochemical reactions involved in the creation of secondary pollutants (e.g. Gramsch et al., 2006; Barraza et al., 2017). Moreover, despite a well developed network of air quality monitoring stations

across the country, the spatial and temporal density of the data does not allow for a detailed observation-based study of atmospheric pollutants transport. As a result, chemistry-transport modeling offers a solution to cope with this issue and provide insights regarding the magnitude and mechanisms of advection of pollutants at the regional scale.

The present work studies, for one summer month and one winter month in 2015 in central Chile, through chemistry-transport simulations with WRF-CHIMERE, (i) the contribution of pollutants emitted in Santiago to the regional atmospheric composition, (ii) the reciprocal contribution of regional emissions to air pollution in the capital city basin, (iii) the corresponding 3-dimensional advection patterns of particulate matter and ozone. The methodology and data are described in Section 2, the relative contributions of transport and the underlying advection processes are presented in Section 3. These results are discussed in Section 4 and conclusions are gathered in Section 5.

## 2 Data and methods

### 2.1 Modeling setup

The chemistry-transport simulations are performed with the Weather Research and Forecasting (WRF) mesoscale numerical weather model from the US National Center for Atmospheric Research (Skamarock et al., 2008) to simulate the meteorological fields, and CHIMERE to compute chemistry and transport (Mailler et al., 2017). Anthropogenic emissions are based on the EDGAR HTAP V2 inventory (Janssens-Maenhout et al., 2015). The simulation domain has a 5 km spatial resolution, extending over 200 latitudinal and 100 longitudinal grid points, and is centered on Santiago (white domain CHILE_5K in Figure 1a). The parameterizations and model configuration used for WRF are presented in Table A1. WRF is applied to 60 vertical levels up to the highest elevation of 50 hPa. Initial and boundary conditions rely on the NCEP FNL analysis data sets, with a 1° by 1° spatial resolution and 6-hour temporal resolution, from the Global Forecast System (NCEP, 2000). Land-use and orography are extracted from the modified IGBP MODIS 20-category database with 30 sec resolution (Friedl et al., 2010). CHIMERE is a Eulerian 3-dimensional regional Chemistry-Transport Model, able to reproduce gas-phase chemistry, aerosols formation, transport and deposition. In this work, the 2017 off-line version of CHIMERE is used (Mailler et al., 2017). The model configuration is described in Table A1, with the same horizontal domain as for WRF, applied on 30 vertical levels up to 150 hPa. Emissions from the EDGAR HTAP V2 inventory are downscaled and split in time down to daily/hourly rates following the methodology of Menut et al. (2013). Biogenic emissions fluxes in CHIMERE are computed online using the MEGAN (Model of Emissions of Gases and Aerosols from Nature) model (Guenther et al., 2006). Emission fluxes from the vegetation are based on air temperature, photosynthetic photon flux density and leaf area index. As an example, an emission map of isoprene ($C_5H_8$, a VOC involved in the formation of $O_3$ and secondary organic aerosols) as computed in CHIMERE for a given day in January 2015 is shown in Figure A1, illustrating the meriodional gradient of vegetation cover.

Two different simulation periods are investigated to account for summertime and wintertime differences in atmospheric composition and advection dynamics. "Summertime" hereafter refers to the simulated period 4 January to 3 February 2015. A 20-days spin-up period between 15 December 2014 and 3 January 2015 is used. "Wintertime" refers to the simulated period between 1 July and 31 July 2015. A spin-up period from 15 June to 30 June 2015 is applied. For each of these periods,

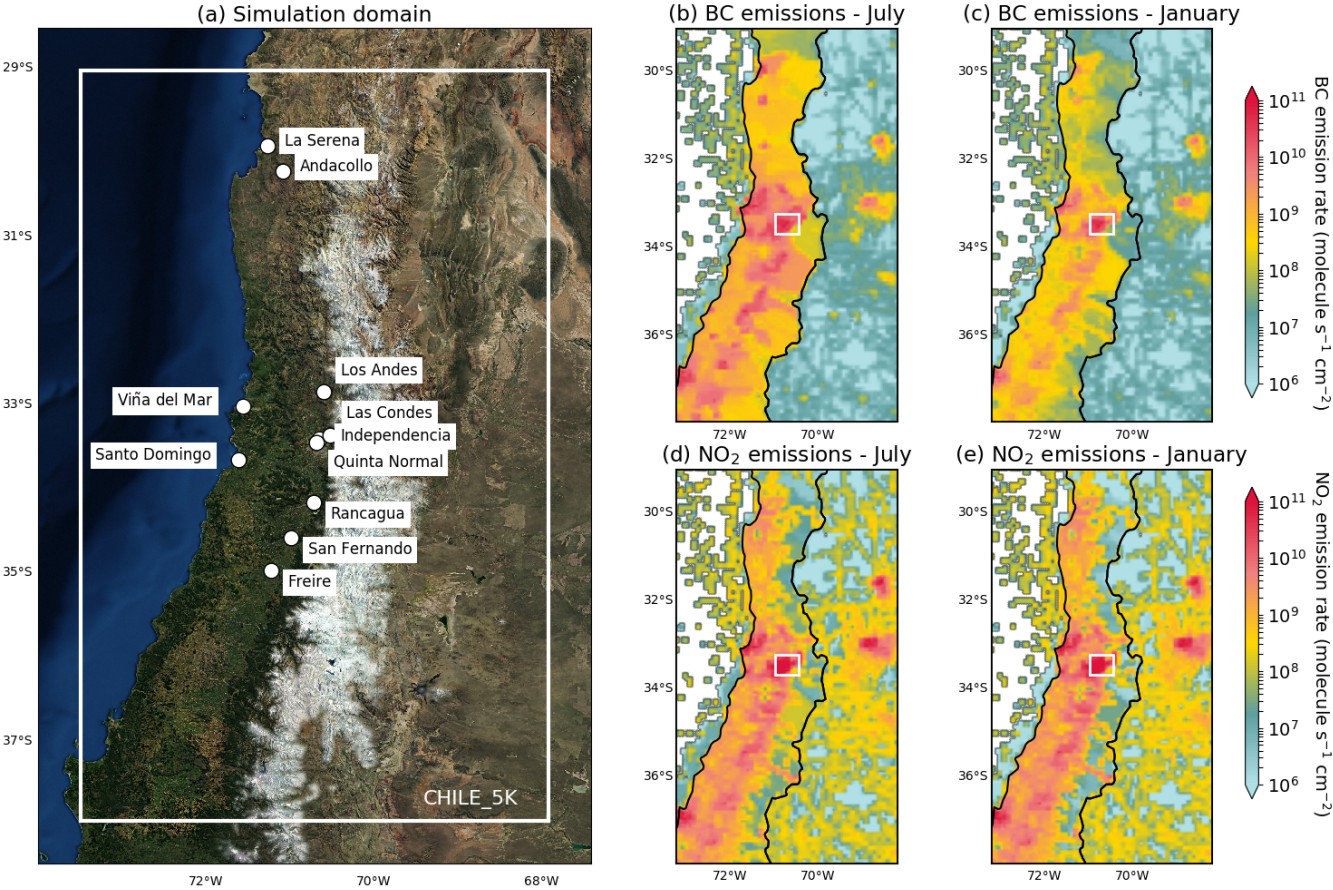

**Figure 1.** (a) Simulation domain at 5 km spatial resolution centered over Santiago. Locations of interest are designated with white dots. Map background layer: Imagery World 2D, ©2009 ESRI, (b) Daily average surface emission rate of black carbon (BC) for July 2015, (c) same as (b) for January 2015, (d) same as (b) for nitrogen dioxide ($NO_2$), (e) same as (d) for January 2015.

two simulations are performed. One uses a full emissions inventory including all emissions within the region, referred to as "baseline" henceforth. The baseline simulation for wintertime is named "WB" and the baseline simulation for summertime is denoted "SB". The second simulation corresponds to the case where the city of Santiago would emit no anthropogenic
pollutants. In this regard, all anthropogenic emissions within the area corresponding to the white rectangle in Figures 1b through 1e are set to zero in the simulation referred to as "contribution", hence the designation "WC" and "SC" for the corresponding wintertime and summertime simulations, respectively. The difference in simulated concentrations between baseline and contribution runs shows the proportion of pollutants transported to and exported from Santiago. Aerosol feedback on meteorology is not taken into account here in order to isolate the sensitivity to emission rates only, so that for a given season
the same WRF fields are used for both emission cases.

Emissions downscaled from the HTAP V2 inventory and input into CHIMERE are shown in Figures 1b through 1e. The seasonality of BC emissions is clear, given the major role played by residential heating which takes place mostly in wintertime. Monthly $NO_2$ emissions are less variable throughout the year, related to the sustained traffic and industrial sources that are nearly constant through time. Figures 1b through 1e also evidence that Santiago city (represented with a white rectangle) features the highest emission rates for both pollutants and dominates the signal over the simulation domain. In the continuation, given their relevance for the associated seasons, $PM_{2.5}$ in wintertime and $O_3$ and its precursors in summertime will be considered as the variables of interest, although $PM_{2.5}$ in summertime and $O_3$ in wintertime can be occasionally discussed. $PM_{2.5}$ include all primary aerosol species (including dust and sea-salt), as well as secondary organic aerosols, but do not incorporate aerosol water.

## 2.2 Simulation validation

Surface meteorology and pollutants concentrations are validated using data from the automated air quality and meteorology monitoring network of Chile, known as Sistema de Información Nacional de Calidad del Aire (https://sinca.mma.gob.cl/index. php/, last access October 1 2020). Different stations across central Chile are considered depending on data availability for the simulated periods (stations locations can be found in Figure 1a). Meteorological vertical profiles in downtown Santiago, for a few days in July 2015, were provided by the Chilean meteorological office, Dirección Meteorológica de Chile.

Simulation scores for surface and vertical profile meteorology are gathered in Tables A2 and A3. Biases on daily mean temperature range between -1.23 °C and 0.31 °C in wintertime except for the mountainous location of Los Andes where the bias reaches -3.33 °C. In summertime the bias is between 0.07 °C to 0.67 °C. For both periods, correlations on surface temperature vary between 0.7 and 0.89, except for Viña del Mar where it drops to 0.25 in wintertime and 0.18 in summertime, which can be explained by the location of this station near the ocean. The corresponding grid point in the model straddles ocean and land hence featuring a strong gradient, and as a result may not be representative of this coastal city. This remark applies to all meteorological variables at coastal sites. The model shows a negative bias on surface relative humidity with average mean biases of -15% to -20% but shows fair correlations around 0.8 to 0.9. Surface winds are fairly reproduced, with limited biases and wind gusts well captured, although the correlations can be low for some locations. Figures A2 and A3 compare observed and simulated surface wind distributions for four sites. Summertime shows a good reproduction of wind regimes for all locations, expect for a small positive bias on speeds. In wintertime, winds are more variable and follow less clear patterns, so that the model performance is not as good, especially for the coastal locations of La Serena and Santo Domingo. A positive bias on speed is also observed. Most features of the vertical profiles of temperature, relative humidity and winds are well reproduced, with very good correlations and limited biases for four winter days in downtown Santiago (Tab. A3). On the whole, these statistics give confidence in the ability of the model to produce realistic transport events for both seasons.

Figure 2 shows a scatter plot of measured and modeled daily mean concentrations of $PM_{2.5}$ in wintertime and daily maximum mixing ratio of $O_3$ in summertime. For $PM_{2.5}$ the simulation performs better for sites far inland with correlations of 0.77 and 0.69 for Rancagua and Independencia, respectively. Correlations for the two coastal sites considered (Viña del Mar and La Serena) are more moderate with 0.34 and 0.25, respectively. The same issue of straddling grid points explained previously leads

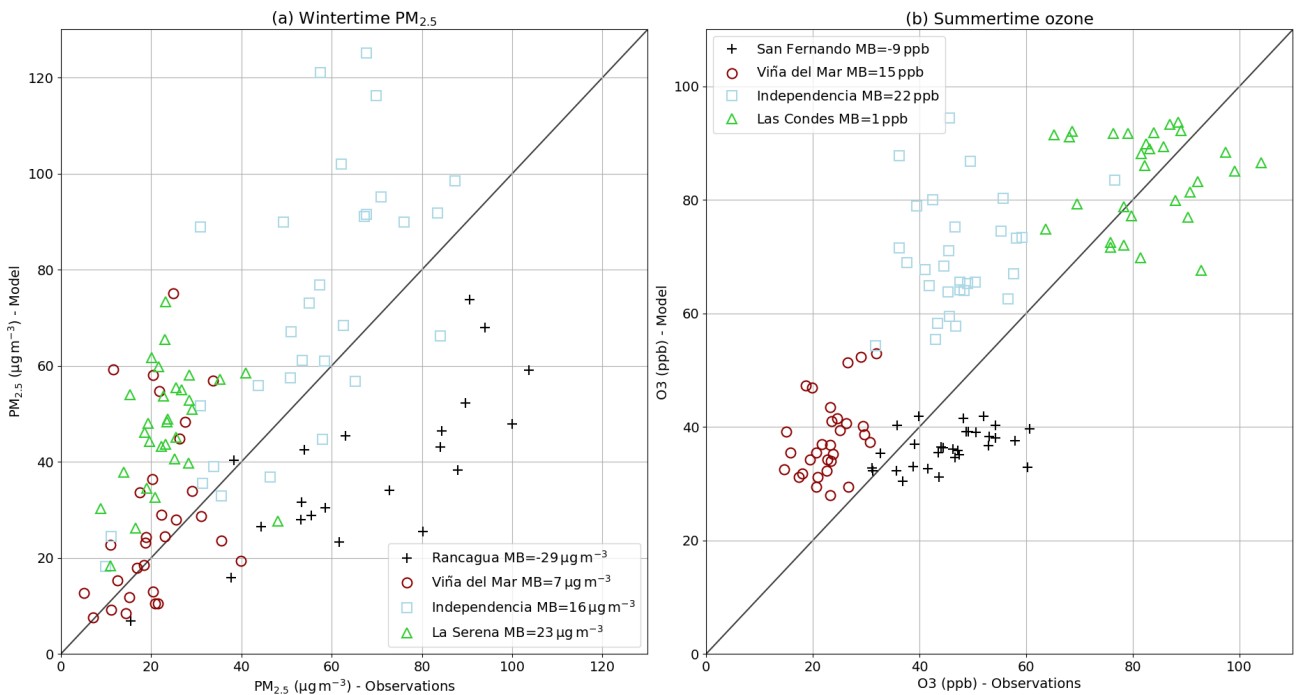

**Figure 2.** Comparison between observation and simulation for (a) wintertime daily average PM$_{2.5}$ concentrations, (b) summertime daily maximum O$_3$ mixing ratio. Wintertime and summertime correspond to the periods defined in Section 2.1, respectively. MB is the mean bias.

to these degraded statistics: while observations at these sites show a chaotic time series, the model produces a smoother diurnal cycle due to the grid point being partially over the ocean. However for those two sites, biases remain small, while the model systematically underestimates concentrations in Rancagua, and slightly overestimates PM$_{2.5}$ in Santiago (Independencia). For O$_3$ the picture is similar with a good reproduction of daily peaks in summertime. Despite the relatively coarse resolution of the simulation and strong spatial heterogeneity in precursors emissions, limited biases of a few ppb are obtained on O$_3$ peaks in

summertime, down to only 1 ppb at the most O$_3$-polluted site of Las Condes (northeastern Santiago). The diurnal cycle of O$_3$ is also well reproduced with hourly correlations (not shown here) of 0.67 for Viña del Mar and 0.8 to 0.9 for the three other sites. In parallel, summertime NO$_x$ mixing ratios within Santiago city (not shown here) are well captured by the model with mean biases between 0.05 and 1.23 ppb for three stations in Santiago (Las Condes, Puente Alto - southeastern Santiago - and Independencia), associated with decent hourly correlations between 0.43 and 0.59.

The lack of available measurements for NO$_x$ and VOC in central Chile hinders the simulation validation regarding these precursors. However, the HTAP inventory has proved reliable for large urban basins in Argentina and Brazil in terms of magnitude of VOC emissions (Puliafito et al., 2017; Dominutti et al., 2020), and more generally all across southern South America for NO$_x$ (Huneeus et al., 2020). Consequently, we postulate that emission rates input in the model are appropriate, hence providing adequate chemical regimes when it comes to the simulation of O$_3$ concentrations. Besides, the known biases

of HTAP on these pollutants are critical when it comes to more detailed approaches for policy making but for the purpose of

the present work, having the proper total amount is sufficient as we apply our own downscaling methodology, do not discuss very high-resolution processes, and rely mostly on sensitivity analysis.

In conclusion, $PM_{2.5}$ in wintertime and $O_3$ and its precursors in summertime, the key pollutants for their respective seasons, as well as meteorological conditions, are fairly reproduced by the model for a selection of sites throughout central Chile, which gives confidence in the model outputs that are described and analyzed in the following sections.

## 3 Results

Hereafter are described well-known general meteorological features generated by the model, that provide a first clue regarding the advection of polluted air masses in the region, and constitute a reference frame accounting for the results described in the continuation.

The semi-permanent South Pacific High, centered around ($30°$S, $110°$W), along with the elevated Andes cordillera, are two large-scale drivers of the surface wind systems in central Chile. The persistent high induces high velocity southwesterlies blowing along the coast during both daytime and nighttime (Fig. 3). These also penetrate deep into land as far as the Andes in summertime during the day, before being blocked by the mountains. On the other side of the Andes, less intensive easterlies coming from Argentina encounter the foothills. Also, the presence of the Andes leads to the development of mountain-valley circulation patterns (e.g. Whiteman, 2000) when the differential heating between narrow valleys and wide plains at the onset and offset of the day lead to the creation of upslope westerlies during daytime (as seen in Fig. 3b), and a reversal at nighttime with downslope easterlies (Fig. 3d). Although this pattern can be perturbed by clouds or synoptic-scale transient phenomena such as coastal lows, it represents the typical surface wind diurnal cycle for basins along the Andes. From these mean wind fields, the dominant advection pathways of pollutants can be inferred. Polluted air masses are on average blown towards the Andes and the north during daytime in summertime, and have more complex dynamics in wintertime but also transport northward in general. Deeper and more turbulent planetary boundary layer heights during daytime, as observed in summertime (Fig. 3a) also enable the vertical export of pollutants up into the free troposphere (FT) where they can be advected farther, while wintertime shallower boundary layers (Fig. 3b) imply more stagnation of air masses.

### 3.1 Impact of emissions from Santiago on regional atmospheric composition

### 3.1.1 Wintertime $PM_{2.5}$

Consistently with the mean wind fields and emission rates of pollutants in central Chile discussed previously, anthropogenic emissions in the city of Santiago significantly influence surface atmospheric composition over a large region, over land, the Andes and the Pacific ocean. The following results are based on the analysis of sensitivity to emissions from Santiago described in Section 2.1: the difference between the simulation with (baseline case) and without (contribution case) emissions from Santiago yields their contribution to atmospheric composition over the domain. Figures 4a and 4b show the average wintertime $PM_{2.5}$ plume (absolute and relative, respectively) attributable to emissions from the capital city area. The direct western vicinity

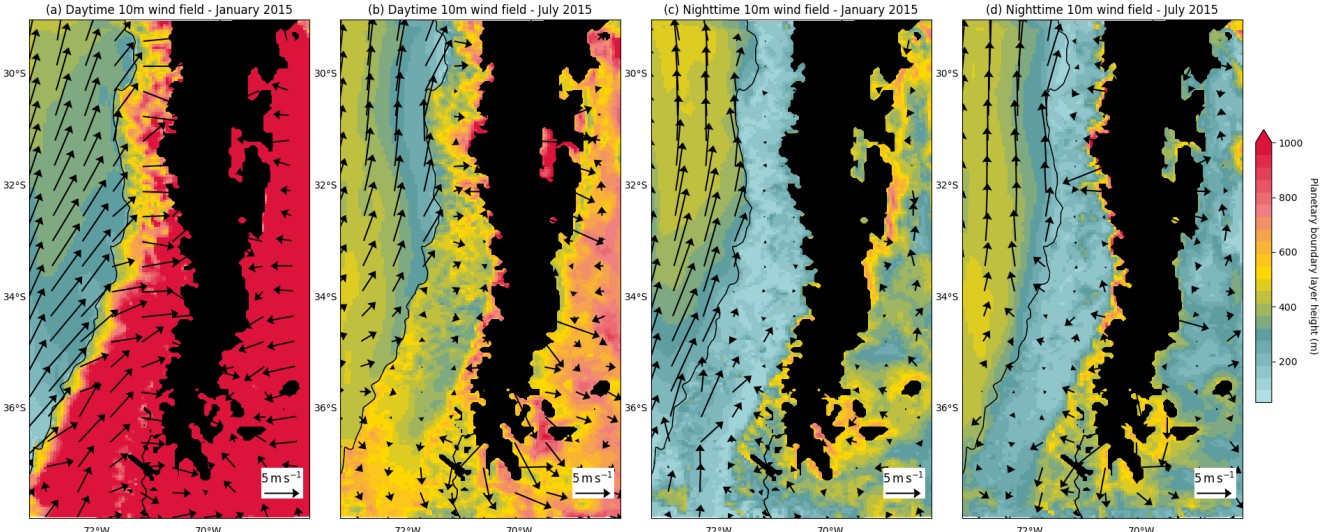

**Figure 3.** Average 10 m wind field (arrows) and planetary boundary layer height (colormap) simulated by WRF (a) during daytime in January 2015, (b) during daytime in July 2015, (c) during nighttime in January 2015, (d) during nighttime in July 2015. Black areas show grid points with elevation in excess of 2000 m a.s.l.

of the Santiago basin receives, on average 5 $\mu$g m$^{-3}$ to 15 $\mu$g m$^{-3}$ coming from the capital city, a few tens of kilometers from the source, corresponding to more than 30% of the signal simulated in the baseline scenario for this area. At the scale of hundreds of kilometers, the export drops to a few $\mu$g m$^{-3}$, corresponding to 5% to 20%. It is worth noting that the relative
contribution of emissions from Santiago remains greater than 5% on a area as large as more than 8° meridionally and 3° zonally, hence stressing the significant impact of the capital city on atmospheric composition for the whole region (Fig. 4b). In particular, the southern part of the plume is transported over the Andes down to Argentina with a large spread of several degrees of longitude, whereas the northern part extends mostly along the coast in a narrower manner and transports as far as the boundary of the simulation domain.

More specifically, urban areas along the north-south axis of Santiago (Curicó, San Fernando, Rancagua and Los Andes in Fig. 4a) receive 1 $\mu$g m$^{-3}$ to 2 $\mu$g m$^{-3}$ from Santiago on a hourly basis on average, corresponding to 4% to 8% of the baseline concentrations (Fig. 4c). Sporadically, up to more than 20 $\mu$g m$^{-3}$ in Rancagua and 9 $\mu$g m$^{-3}$ in San Fernando, Curicó and Los Andes can be attributed to emissions from Santiago. These significant contributions likely lead to alert thresholds crossing for some hours in these cities.

On the eastern side of the Santiago basin is the Andes cordillera. We examine the contribution of Santiago emissions in a mountain locality (San Gabriel, 1250 m a.s.l.) and a summit (Maipo volcano, 5264 m a.s.l.) along the Maipo canyon, southeast of Santiago. We find that for San Gabriel 34% (1 $\mu$g m$^{-3}$) of PM$_{2.5}$, on average, is transported from the urban basin. This is consistent with the mountain-valley circulation patterns aforementioned, leading to the intrusion of urban air masses deep into the canyon (Lapere et al., 2021b). We acknowledge that the estimate of 34% is probably larger than reality since the HTAP

inventory does not capture properly local emissions in the village of San Gabriel, which likely dominate the signal, especially with wood burning for residential heating being largely used in such villages in wintertime. In the Maipo volcano area, a small contribution in absolute value is found (less than $0.5\,\mu\mathrm{g\,m^{-3}}$) although it can reach up to more than $2\,\mu\mathrm{g\,m^{-3}}$ occasionally, but this corresponds to 20% of the signal there on average. This area is covered in snow during wintertime, so that despite the small magnitude of the import of $PM_{2.5}$ it can lead to significant radiative effects when deposited (Rowe et al., 2019), especially given the large fraction of BC in $PM_{2.5}$ emitted in Santiago, around 15% according to the HTAP emissions inventory.

The Viña del Mar-Valparaíso area is the second largest populated region of Chile, located on the coast of central Chile, approximately $100\,\mathrm{km}$ west of Santiago. In wintertime, it is downwind of Santiago, which leads to an average import of particulate matter from the capital city of $3\,\mu\mathrm{g\,m^{-3}}$ (18%) and sporadically up to $18\,\mu\mathrm{g\,m^{-3}}$. Again, air quality in this urban area is worsened by export from Santiago, by a significant share. Further north, at the location of La Serena, which also suffers from bad air quality in wintertime, the contribution of Santiago emissions is more moderate but still remains significant in absolute value although its contribution is only 1%.

### 3.1.2 Summertime $O_3$

In summertime, except for $PM_{2.5}$ emitted by biomass burning events (not considered here), $O_3$ is the pollutant raising concern. Combined significant emissions of $NO_x$ and VOC are required, in the presence of sunlight, to generate high mixing ratios of $O_3$. However, a lot of non-linearities are involved in the tropospheric $O_3$ cycle, so that the sensitivity to its precursors emissions is not straightforward. For instance, imbalances in the ratio of $NO_x$ and VOC decrease $O_3$ formation. Given the crucial role of photolysis in $O_3$ formation, it features a strong diurnal cycle, with levels coming back to low values at night. Thus, the important variable to determine whether $O_3$ pollution is high is its daily maximum mixing ratio, on which we will mostly focus hereafter.

Figure 5a shows the average of maximum hourly $O_3$ mixing ratio at ground-level observed each day in the baseline case (SB). In the baseline scenario, $O_3$ is mostly found at harmful levels in the vicinity of Santiago, and mainly on its eastern side. Again, mountain-valley circulation accounts for this observation: afternoon westerlies blow $O_3$ precursors, present in large amounts in urban air masses, towards the Andes. $NO_x$ lifetime is a few hours at most, depending on the reactivity of VOC and $NO_2$ density (e.g. Laughner and Cohen, 2019), while most VOC have an atmospheric lifetime of several days (e.g. Monod et al., 2001), so that on the way, $NO_x$ are more consumed than VOC. Consequently, while urban air is mostly a $NO_x$-rich environment, precursors ratios become more balanced along with the export so as to create more favorable conditions for $O_3$ formation once reaching less urbanized areas. Such a mechanism is observed for Paris and its suburbs for instance (e.g. Menut et al., 2000). Export by easterlies occur less frequently and mostly at night when $O_3$ cannot be created due to lack of sunlight, which is why the rural area west of Santiago shows smaller $O_3$ maxima. This mechanism explains the large concentrations of $O_3$ mostly found east of Santiago. Except for the center part of the domain where average daily maxima reach more than 100 ppb, $O_3$ pollution is less concerning elsewhere in central Chile where they range between 10 ppb and 35 ppb.

Figure 5b shows the decrease in $O_3$ daily maxima induced by eliminating emissions of the Santiago basin. The spatial pattern is again consistent with the mechanism introduced previously: emissions of precursors from Santiago are the main origin for

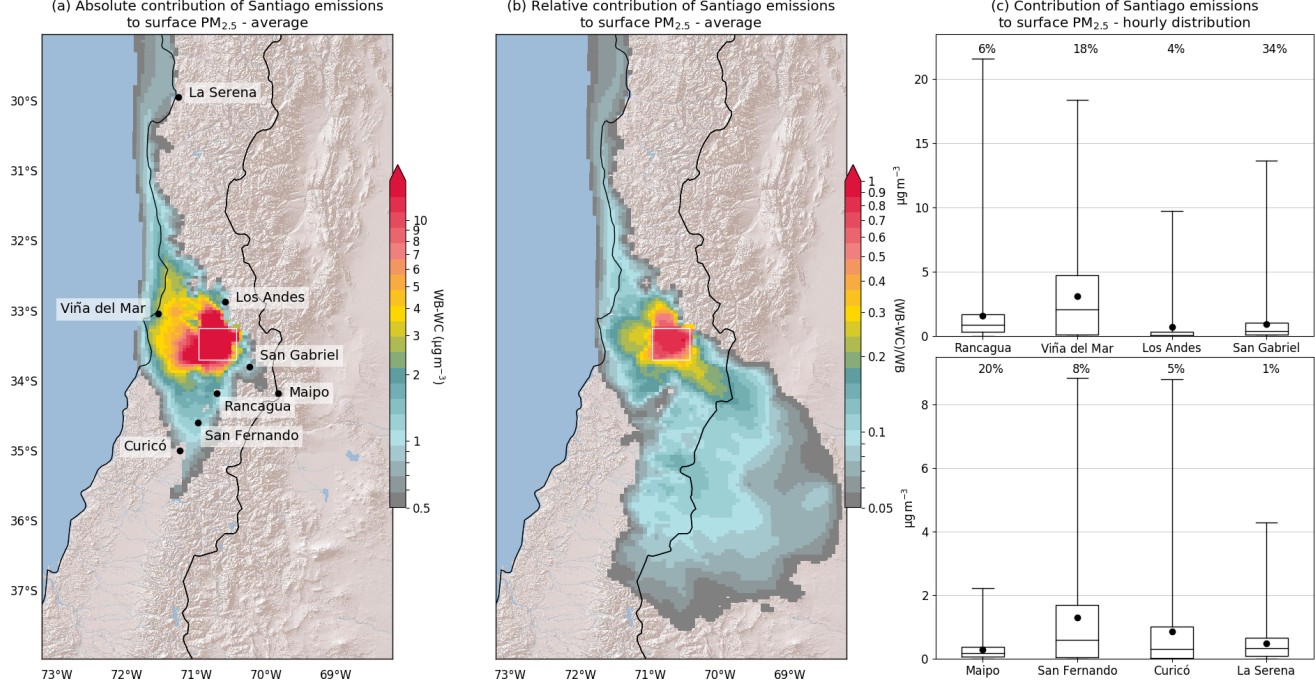

**Figure 4.** (a) Average ground-level PM$_{2.5}$ concentration difference between WB (winter baseline case) and WC (winter contribution case) yielding the Santiago emissions contribution. Concentrations in excess of $0.5\,\mu g\,m^{-3}$ are displayed. White rectangle shows the area where emissions are set to zero in WC. Map background layer: World Shaded Relief, ©2009 ESRI, (b) same as (a) in relative contribution i.e. (WB-WC)/WB. Contributions in excess of 5% are displayed. (c) Distribution of Santiago emissions contribution to hourly PM$_{2.5}$ concentration at several sites across central Chile - horizontal lines are quartiles, whiskers show minimum and maximum, dots are the means. All figures are based on hourly concentrations and are for wintertime 2015. In panels (a) and (b) only grid points where the distribution of concentrations in scenario WC is different than in scenario WB at the 90% level, based on a t-test, are shown.

O$_3$ formation in the Andes and north of the city, with a reduction of more than 50 ppb of the daily maxima over this area when the capital city no longer emits pollutants.

More specifically, the northern city of Los Andes shows a decrease of its daily maxima by 15 ppb on average (Fig. 5b) while the average mixing ratio drops from 40 pbb to 33 ppb (Fig. 5c). Similarly, at the ski resort site of Valle Nevado (3000 m a.s.l.), which shows concerning levels of more than 60 ppb of O$_3$ on average in the baseline case, the mixing ratio drops to an almost constant value of 33 pbb and does not go above 43 ppb. This points to O$_3$ in Valle Nevado being almost exclusively attributable to Santiago emissions. To a lesser extent, the location of San Gabriel shows the same trend although the still wide distribution in scenario SC advocates for a significant contribution of local sources as well. The impact of Santiago emissions in the vicinity of the Maipo summit is small but remains significant with a decrease by 4 ppb of the daily maxima and average mixing ratios. Given the nature of the locations Valle Nevado and Maipo, the narrow distribution of O$_3$ hourly mixing ratio, averaging at 33 ppb and ranging between 25 ppb and 40 ppb, recovered in scenario SC for those sites is indicative of the

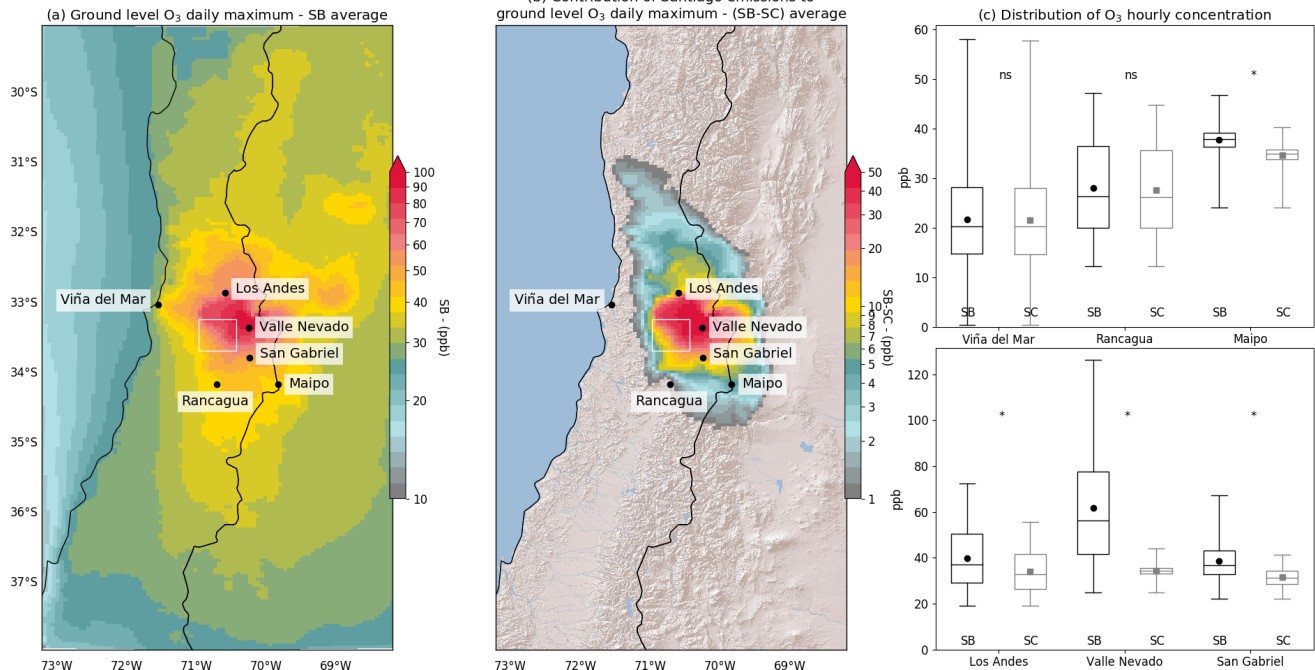

**Figure 5.** (a) Ground-level $O_3$ average daily maximum of hourly mixing ratio in scenario SB, (b) Difference between SB and SC in ground-level $O_3$ average daily maximum of hourly mixing ratio. Map background layer: World Shaded Relief, ©2009 ESRI, (c) Distribution of hourly $O_3$ mixing ratio at six locations in scenario SB (black) and SC (gray) - horizontal lines are quartiles, whiskers show minimum and maximum, dots show the average. "ns" indicates that the distribution in the SB and SC scenarios cannot be distinguished at the 90% level based on a t-test. "*" indicates that they are different at the 99% level based on that same t-test.

background concentration for the region, i.e. the distribution of $O_3$ mixing ratio that would be observed at a site not influenced by anthropogenic emissions.

On the other hand, the western and southern areas adjacent to Santiago are barely sensitive to its emissions. In Figure 5c, the distribution of hourly $O_3$ mixing ratio at Viña del Mar and Rancagua is nearly the same in both scenarios, except for the maximum at Rancagua which is a few ppb lesser in scenario SC than SB. For those two locations, the difference in $O_3$ distribution with or without Santiago emissions is not significant at the 90% level, while for all other locations, distributions are significantly different at the 99% level.

## 3.2   Contribution of regional emissions to atmospheric composition in Santiago

Similarly to the previous approach, we can, in a symmetric manner, deduce the contribution of transport from remote sources to atmospheric composition in Santiago city. This is achieved by looking at concentrations in the contribution case, that exclusively originate from non-local sources. Hereafter the analysis focuses again on $PM_{2.5}$ in wintertime and $O_3$ in summertime.

### 3.2.1 Wintertime PM$_{2.5}$

Figure 4b indicated that local emissions largely dominate the wintertime PM$_{2.5}$ signal for Santiago, with 50% to 100% of the mean surface concentration of PM$_{2.5}$ originating locally within the white rectangle. Nevertheless, the contribution of transport is also significant although heterogeneous in Santiago. Figure 6a shows that in the baseline scenario, the northern and western

parts of Santiago feature higher levels of PM$_{2.5}$, with average concentrations ranging between $30\,\mu\mathrm{g}\,\mathrm{m}^{-3}$ and $100\,\mu\mathrm{g}\,\mathrm{m}^{-3}$ for the whole metropolis. In the contribution scenario (Fig. 6b), this pattern is partly recovered, with a smoother gradient from west to east. The underlying conclusion is twofold. First, the transport of pollutants in Santiago mostly comes from the west, which is consistent with the presence of the mountain range in the east not featuring many sources of pollution, and the dominant westerly daytime wind direction in wintertime. Second, the districts of Santiago facing the worst air quality are also the ones

where the transport of pollutants is larger.

However, the differing patterns between Fig. 6a and 6b also shows that local sources in these districts are also stronger. If emissions were similar, the observed gradient in WB would be closer to that in WC. Consistently with this observed westward gradient, we define 5 zones of interest, comprising 6 grid points each along a meridional axis (rectangles and dots in Fig. 6b). This arrangement ensures that most of the city is covered while maintaining the west-east variability. For each zone we look

at the distribution of PM$_{2.5}$, averaged over the 6 grid points, in scenarios WB and WB-WC (Fig. 6c) and WC (Fig. 6d). When averaged meridionally, the westward gradient is also obtained in scenario WB, and conserved when the contribution of transport is substracted (WB versus (WB-WC) in Fig. 6c). On average, the contribution of transport to PM$_{2.5}$ concentration ranges between $10\,\mu\mathrm{g}\,\mathrm{m}^{-3}$ for the westernmost area and $5\,\mu\mathrm{g}\,\mathrm{m}^{-3}$ for the easternmost part, with a monotonic spatial variation (Fig. 6d). However, this amount always corresponds to 13% to 15% of the WB concentrations. This number is well in line

with Barraza et al. (2017) that found 9% of PM$_{2.5}$ in Santiago coming from coastal sources, for the period 2011-2012. It is worth noting that given the observed westward gradient, we also recover that coastal sources are likely the main contributor to imported PM$_{2.5}$. Again, transport is larger in western Santiago, and can sporadically reach up to $30\,\mu\mathrm{g}\,\mathrm{m}^{-3}$, but does not constitute a greater share than in the east.

This averaged picture provides a first clue as to the main origin of PM$_{2.5}$ transport of Santiago but the picture can be

refined by looking at the joint distribution of hourly wind direction and PM$_{2.5}$ concentrations as shown in Fig. 7. At the selected southeastern location, mostly clean air comes from the east, i.e. from the Andes where pollutants sources are scarce (less than $5\,\mu\mathrm{g}\,\mathrm{m}^{-3}$ for almost every hour), while winds blowing from the southwest can transport concentrations as high as more than $20\,\mu\mathrm{g}\,\mathrm{m}^{-3}$ for some hours, pointing to the southern cities of Rancagua (34.2°S, 70.7°W) or San Fernando (34.6°S, 71°W) mentioned previously, or the southwestern urban location of Melipilla (33.6°S, 71.2°W). At the northeastern

site, winds mainly come from the north where only a handful of urban areas are found, hence leading to a transport seldom exceeding $10\,\mu\mathrm{g}\,\mathrm{m}^{-3}$. In the center of the metropolis, winds are either southwesterlies (from the Rancagua and Melipilla areas) or northwesterlies (from the Viña del Mar-Valparaíso area), with both cases leading to similar amounts of imported PM$_{2.5}$ mostly above $10\,\mu\mathrm{g}\,\mathrm{m}^{-3}$, i.e. above average. The picture is similar for the northwestern and southwestern sites although wind directions are shifted.

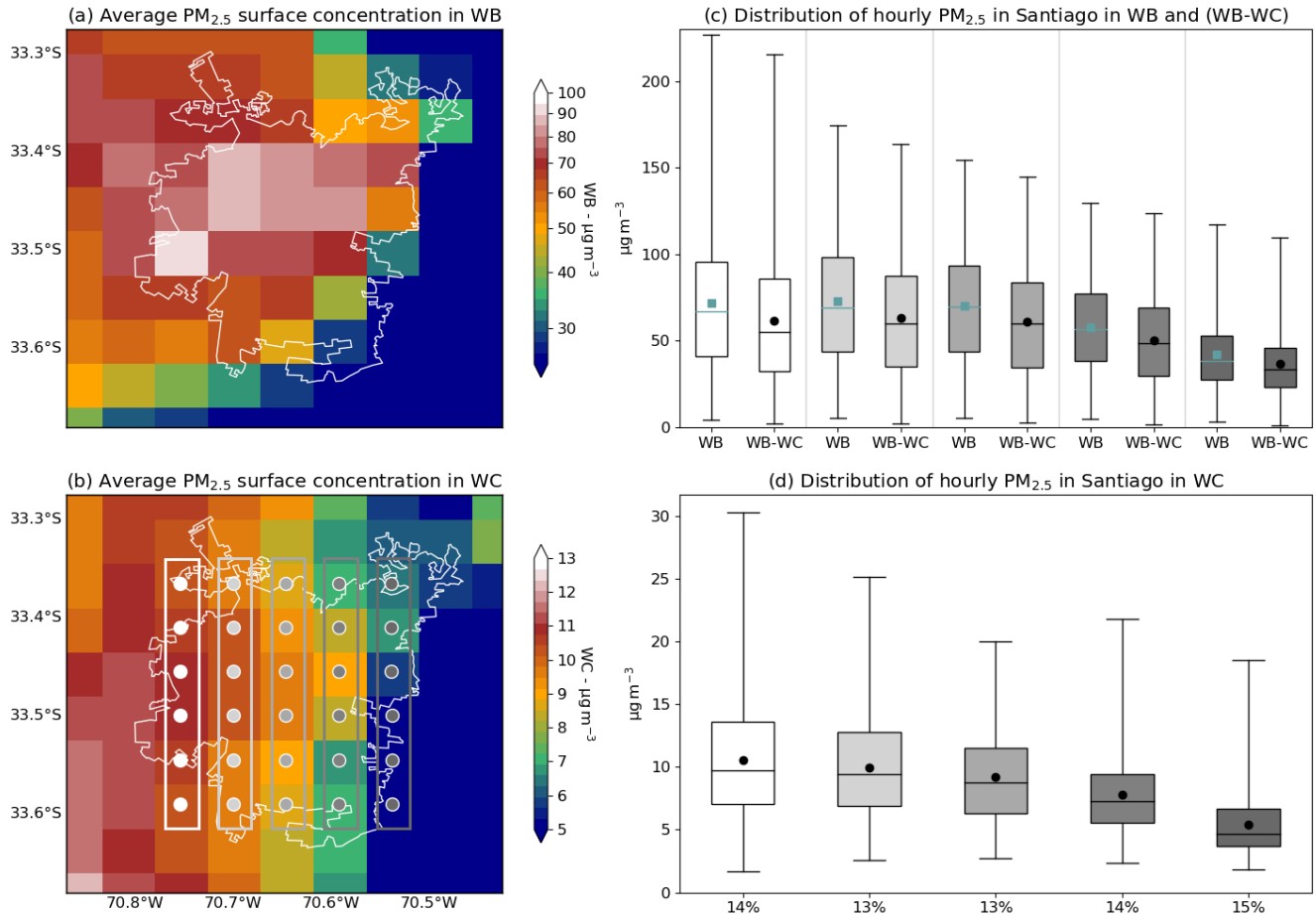

**Figure 6.** (a) Average ground-level $PM_{2.5}$ concentration in scenario WB. White contour shows the boundaries of Santiago city. (b) same as (a) for scenario WC, (c) Distribution of hourly $PM_{2.5}$ surface concentration in WB and (WB-WC). Boxes show the average, median, minimum, maximum, first and third quartiles. Shades of gray correspond to the zones defined in (b) on which an average is made, (d) same as (c) for scenario WC. Percentages indicate the relative average contribution of transport.

In the northwest, center and northeast, the dominant wind directions also coincide with higher maximum relative contributions of transport over the period along these directions (black diamonds in Fig. 7). Sporadically, significant transport events can also come from less frequently observed directions. The maximum relative contribution obtained for the northwest point when winds are from NNE is 75% for example, while such winds occur less than 1% of the time. For southwest and southeast locations, these maximum relative transport episodes are observed when winds blow from the south, in particular in the southeast where it can reach up to 100%.

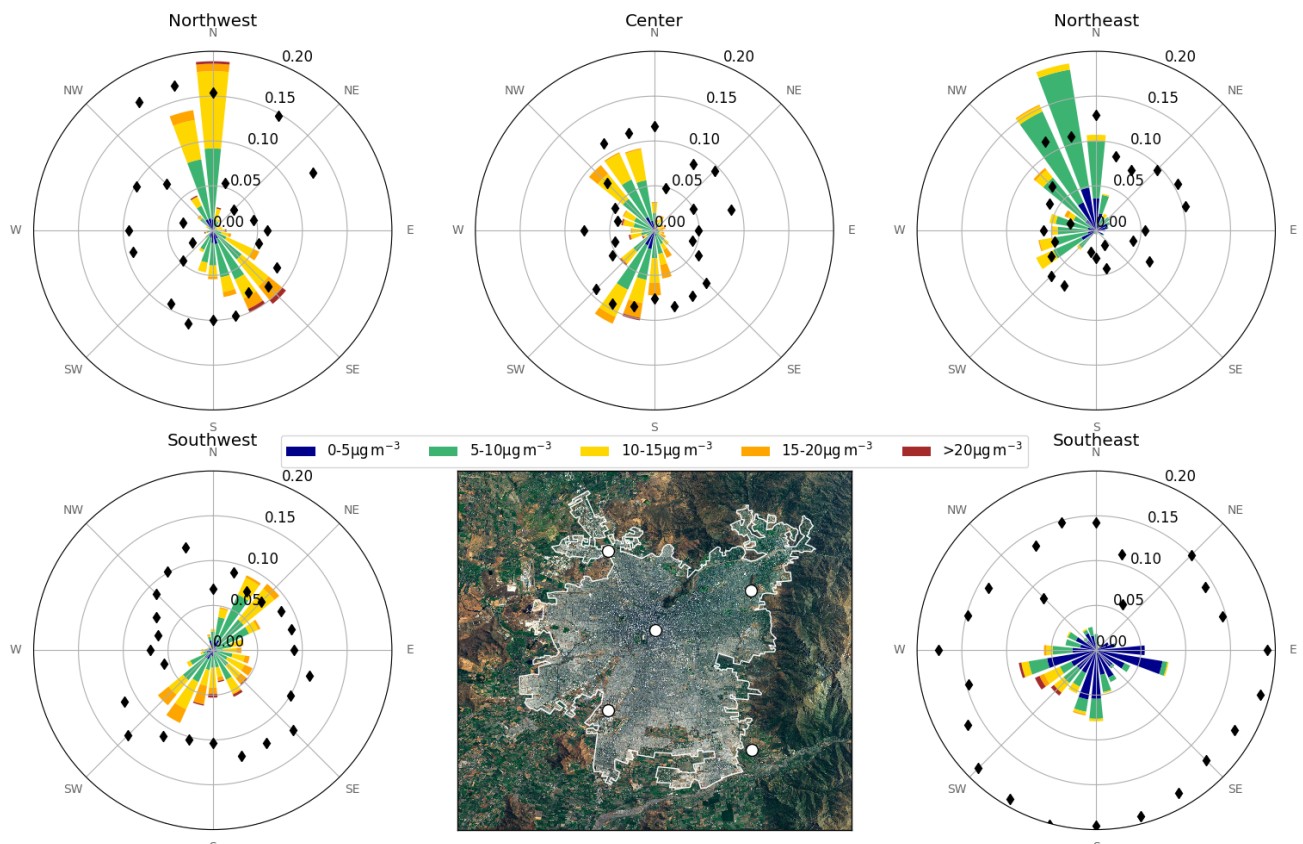

**Figure 7.** Joint distribution of hourly wind direction (length of bar gives the frequency of the corresponding wind direction) and PM$_{2.5}$ concentration (colormap) in scenario WC for 5 locations in Santiago. Black diamonds show the maximum relative contribution of transport (i.e. WC/WB) for each wind direction bin (NB: the scale is the wind frequency scale multiplied by 5, i.e. 0.20 is actually 1.0). The bottom center map shows the location of the considered grid points. Map background layer: Imagery World 2D, ©2009 ESRI.

In summary, wintertime PM$_{2.5}$ concentrations in Santiago are significantly ($5\,\mu g\,m^{-3}$ to $10\,\mu g\,m^{-3}$ on average) and always (at least $1\,\mu g\,m^{-3}$ for every hour) affected by transport, with identifiable origins, and although the different districts are not equally affected in absolute value, the relative burden of imported particulate matter is equivalent.

### 3.2.2 Summertime O$_3$

Based on the same approach, we find that the summertime transport of NO$_x$ within the Santiago basin never exceeds 0.5 ppb, while average values in the SB scenario are between 5 ppb and 40 ppb, with a similar westward gradient as observed for PM$_{2.5}$ (not shown here). At the 90% level, NO$_x$ transport is thus not significant. Similarly, the transport of VOC is homogeneous over the whole basin at 2 ppb on average, while between 20 ppb and 60 ppb in the SB scenario, which is again not significant at the 90% level. Santiago is thus not affected by the transport of O$_3$ precursors.

However, the picture within Santiago city in SB and SC scenarios is complex. In the baseline scenario, the eastern area of Santiago is more affected by $O_3$ pollution compared to the western area (Fig. 8a and 8c), consistently with observations and the literature (e.g. Menares et al., 2020), due to a more balanced VOC/$NO_x$ ratio than in the western area. At Independencia (center of Santiago), the VOC/$NO_x$ ratio at emission is between 1:1 and 2:1 on average. Contrarily, at Las Condes (eastern Santiago) the VOC/$NO_x$ ratio at emission is around 6:1 on average, in the baseline case (not shown here). A typical $O_3$ formation ridge line of the VOC/$NO_x$ concentration ratio in urban areas is around 6:1 to 8:1 (e.g. National Research Council, 1991; Sillman, 1999), so that Independencia features a VOC-limited regime, while Las Condes features a balanced regime favorable to $O_3$ formation, hence the larger amounts found at the latter location.

Figure 8b shows the consequences, on $O_3$ surface mixing ratio, of eliminating emissions within Santiago. Given the configuration described previously, in the baseline case, the VOC-limited districts of western Santiago feature mixing ratios well below the background level due to the titration of $O_3$ by excess quantities of $NO_x$, while the eastern districts feature mixing ratios above the atmospheric background level due to excess $O_3$ formation under a favorable regime. As a result of shutting off emissions, given that there is no import of precursors as evidenced above, the whole area is set to the background $O_3$ level of around 30 ppb described in Section 3.1.2, since there is no influence of anthropogenic pollutants anymore. Therefore, this corresponds to an increase of $O_3$ in western Santiago and a decrease of $O_3$ in eastern Santiago, thus explaining the dipole obtained in Figure 8b. Such an evolution is also clear in the evolution of the distribution of hourly $O_3$ mixing ratios across the city between scenario SB and SC (Fig. 8c). While in scenario SB the distribution is shifted towards larger mixing ratios when going eastward, in scenario SC all distributions are equal (significant at the 99% level). The leveling of mixing ratios, with no gradient across the city in scenario SC constitutes an additional evidence that $O_3$ in Santiago is not affected by long-range transport, otherwise heterogeneous patterns similar to what is observed in Figure 6b would be obtained.

## 3.3 Advection processes

As discussed around Figure 3 and observed in Figure 4 and Figure 5, advection patterns of pollutants differ between wintertime and summertime. So far, the analyses focused on surface fields, but processes along the vertical drive these differences. Figure 9 shows an average latitude/altitude transect, along central Chile, of winds, afternoon mixing layer height and pollutants concentrations for the corresponding season, in the baseline scenario and the Santiago isolated contribution case.

In wintertime (Fig. 9b and 9c), the boundary layer (solid white line) is shallow, and average winds in the FT are strong, consistently with the observed semi-permanent inversion layer in the region. As a result, the $PM_{2.5}$ emitted in large amounts mostly remain trapped within this shallow mixing layer. Injection of polluted air masses into the lower FT can also occur, through mountain venting, as described in Lapere et al. (2021b), explaining why residual concentrations of 1 to 5 $\mu g\,m^{-3}$ are observed higher up in the baseline and contribution scenarios at the latitude of Santiago (Fig. 9b and 9c). Nevertheless, the long-range export of $PM_{2.5}$ from Santiago observed in Figure 4 is mainly driven by advection within the boundary layer, close to the ground, by weak winds, as evidenced by Figure 9c. Except for a shallow residual layer located above the average afternoon boundary layer, pollutants emitted from Santiago remain within it along the transect from Santiago to Viña del Mar. The pattern changes when reaching the seashore however, with significant wind shears lifting the $PM_{2.5}$ layer above the mixing

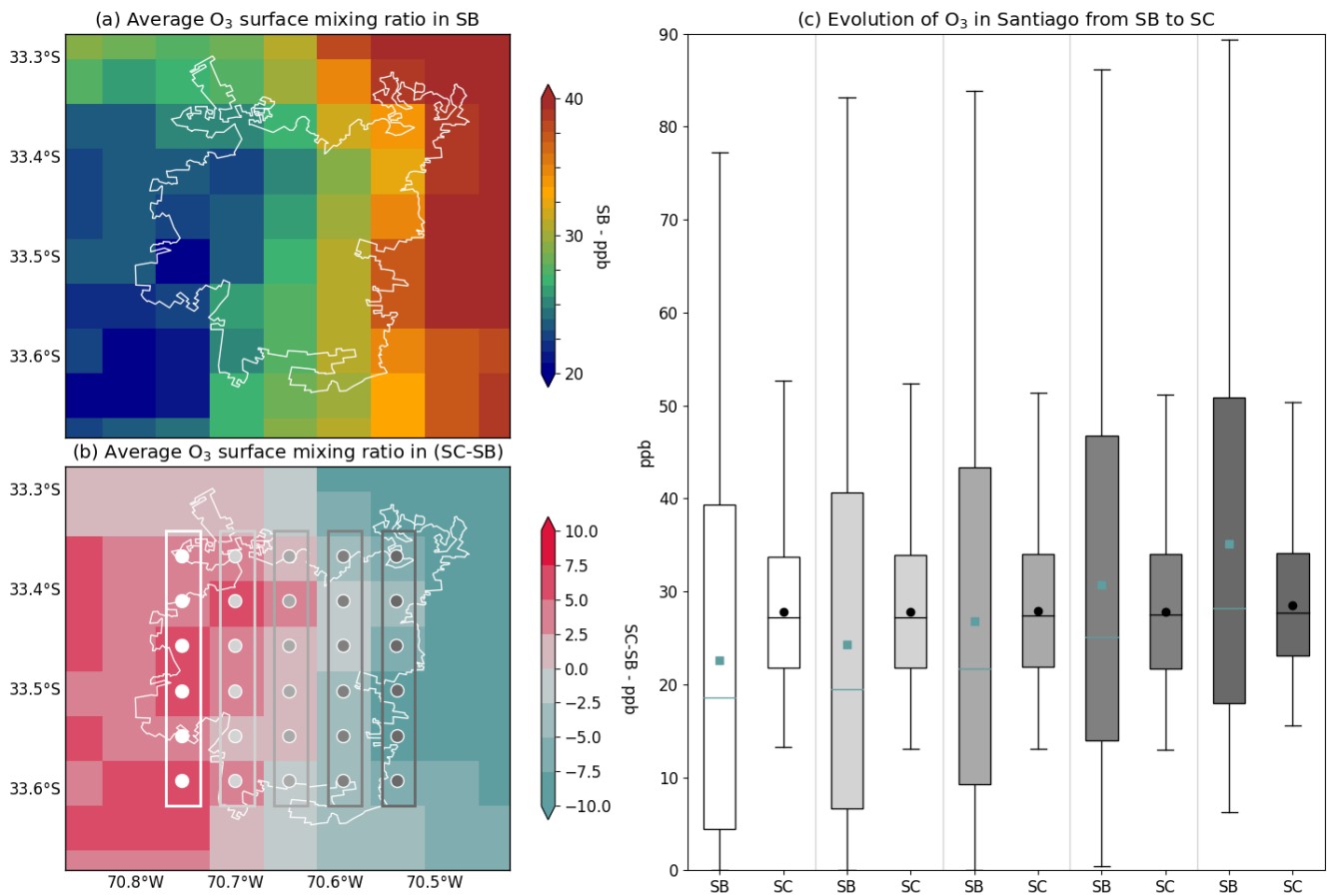

**Figure 8.** (a) Average O$_3$ mixing ratio at ground level in scenario SB in Santiago, (b) same as (a) for scenario (SC-SB), (c) Distribution of hourly O$_3$ surface mixing ratio in scenario SB and SC. Boxes show the average, median, minimum, maximum, first and third quartiles. Shades of gray correspond to the zones defined in (b) on which an average is made.

layer (rightmost part in Fig. 9c) thus explaining the wide northward extent of Santiago contribution, due to transport higher up, by intense southerlies.

In this wintertime averaged picture, the patterns underlying the largest transport events are not obvious. In order to identify these advection patterns, four clusters are designed corresponding to episodes of transport from Santiago to the south (Rancagua), from Santiago to the northwest (Viña del Mar), into Santiago from the south, and into Santiago from the northwest. In the case of transport into Santiago, passive aerosol tracers emitted in the model at the locations corresponding to Rancagua and Viña del Mar are used to discriminate the main direction of origin. The composite of these episodes is defined as the hours when the PM$_{2.5}$ (or tracer) contribution to concentration is greater or equal to its 90$^{\text{th}}$ percentile over the studied period. The meteorological conditions during these particular hours are then compared to the average for the whole period, in order to compute anomalies in the surface wind and pressure fields. Figure A4 shows that southward transport events are associated with

lower than average surface pressure by 1 to 3 hPa and large northerly anomalies over most of the wind field, and in particular in the corridor between Santiago and Rancagua. Conversely, northwestward export is associated with positive anomalies in surface pressure and southerly anomaly in surface winds combined with easterly anomaly in the corridor between Santiago and Viña del Mar. The two aforementioned corridors are evidenced by the topographic contours in Figure A4. These variations are related to the dynamics of the southeast Pacific high, located near (35°S, 110°W): a weakening or southward/westward displacement of the anticyclone lead to the anomalies observed in Figure A4a, while an opposite displacement leads to a situation similar to that of Figure A4b. Consistently, transport events into Santiago are related to symmetrical patterns (not shown here), with transport from the northwest featuring similar anomalies as in Figure A4a and transport from the south originating in the same anomalies as in Figure A4b.

Interestingly, the summertime transect of $O_3$ shows a sharp maximum near the latitude of Santiago, several hundred meters above the ground. Figure 9d shows the formation of an $O_3$ bubble of more than 50 ppb on average, i.e. 15 ppb above the 35 ppb background mixing ratio observed in scenario SB (dominant yellow/green levels in altitude in Figure 9d), around latitude 33°S i.e. slightly north of Santiago, above the planetary boundary layer, extending between 1.5 km and 3 km altitudes. This additional $O_3$ plume is mostly attributable to emissions of precursors in Santiago, that account for more than 15 ppb of $O_3$ on average at the location of the bubble (Fig. 9e), i.e. the background level exceedance. Thus, despite a relatively limited area where precursors from Santiago affect $O_3$ formation near the ground (Fig. 5b), their impact on the vertical is more dramatic. The process underlying the formation of this significant $O_3$ bubble clearly departing from the background, is discussed hereafter. It is also worth noting that even though export of $O_3$ close to the surface is limited (Fig. 5b), it is more widespread higher up, with a residual layer originating from Santiago emissions of a few ppb extending 2 km vertically and transporting northward in the FT along 2° of latitude (Fig. 9e).

Given the proximity of the Andes cordillera to the Santiago basin, the formation of the aforementioned $O_3$ bubble finds its origin in the mountain-valley circulation and the associated mountain venting mechanism. Daytime upslope winds, strong in summertime, lift polluted air masses from the atmospheric boundary layer (ABL) over Santiago into the lower FT, possibly above another location depending on the FT winds direction. McKendry and Lundgren (2000) and Lu and Turco (1996) find that this process is a net sink for boundary layer $O_3$ in British Columbia and the Los Angeles basin, respectively. Henne et al. (2005) find that the effect of venting in an Alpine environment on FT $O_3$ concentrations strongly depends on initial mixing ratios within the vented ABL, with either net production if ABL mixing ratios of $O_3$ are high (urban valleys), or net loss if they are low (remote valleys). However, our study case falls into none of the aforementioned. Such a bubble of $O_3$, detached from the ground, with a mixing ratio much higher than at ground level is not found in the literature to our knowledge. More moderate injections, with $O_3$ mixing ratios lower to similar to surface levels are usually observed. In our case, we find that the venting of precursors from the Santiago polluted ABL leads to the net production of large quantities of $O_3$ in the FT, larger than at the surface. Schematically, there is a larger export of VOC than $NO_x$ in the FT (Fig. A5a and A5b) which makes for a balanced chemical regime (below 0.5 is $NO_x$-limited, above 0.5 is $NO_x$-rich) at around 0.5 at 2 km altitude, while the regime is close to 1 near the surface hence unfavorable to $O_3$ production, along the whole transect, due to dominant urban $NO_x$ emissions

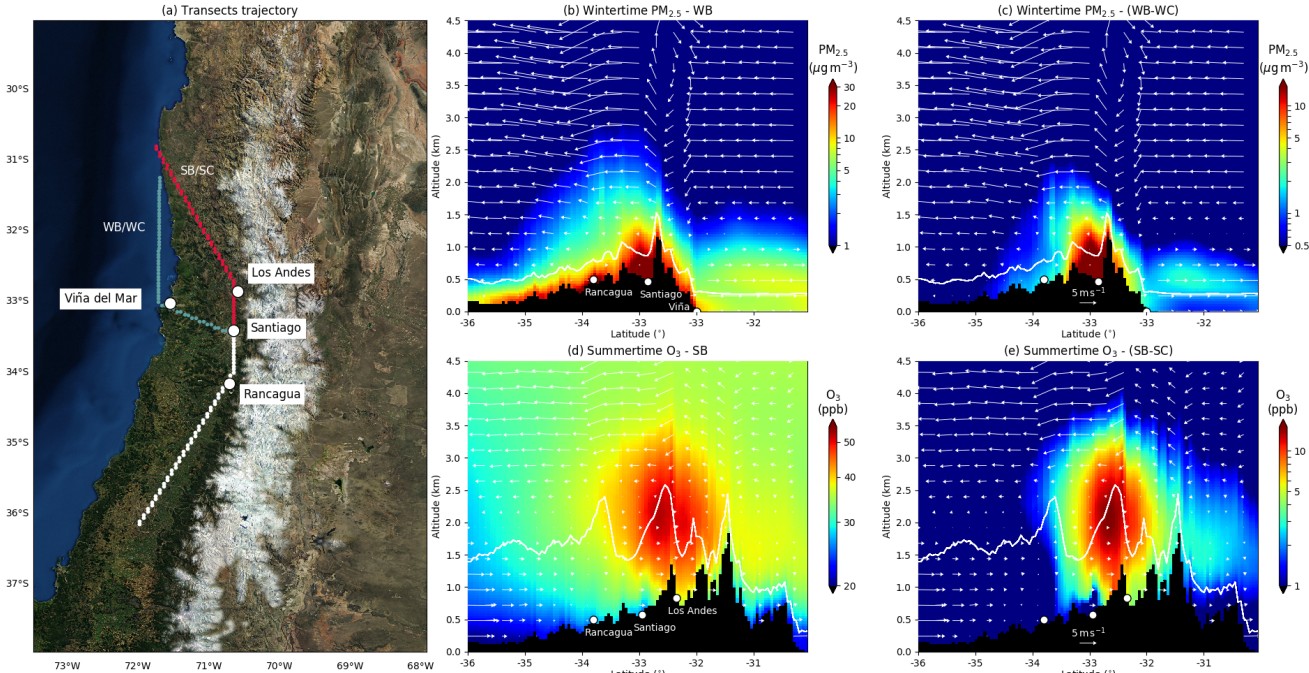

**Figure 9.** (a) Trajectory of the considered latitude/altitude transects and main locations along the way. White dotted line up to Santiago is common for both seasons, blue dotted line is for wintertime, red dotted line for summertime. Values in the transects are not zonally averaged, they correspond to the grid points represented with dots. Map background layer: Imagery World 2D, ©2009 ESRI, (b) Simulated $PM_{2.5}$ concentrations (colormap), wind (white arrows), afternoon boundary layer (white solid line) and terrain elevation (black area), along the white/blue transect. Average for wintertime 2015 - WB scenario. (c) same as (b) for scenario (WB-WC), (d) Simulated $O_3$ mixing ratios (colormap), wind (white arrows), afternoon boundary layer (white solid line) and terrain elevation (black area), along the white/red transect. Average for summertime 2015 - SB scenario. (e) same as (d) for scenario (SB-SC).

(Fig. A5c). Also, the export of Peroxyacetyl nitrate (PAN), which is a $NO_x$ carrier, into the FT, contributes to enhanced $O_3$ formation (Fig. A5d).

Figure 10 further sheds light on the dynamics of this mechanism. Early in the day, at 11:00 UTC (Chile is UTC-3 in summertime), both $NO_x$ and NMVOC (non-methane VOC) are highly concentrated near the ground (morning peak of emissions from traffic), and start ascending the Andean foothills, but no $O_3$ is formed in the region given that sunlight is not intense yet (leftmost column in Fig. 10). A few hours later, at 12:00 LT (15:00 UTC), the precursors have a similar distribution as in early morning, but the photolysis starts taking place and $O_3$ is created at the top of the precursors plume, well above the ground.

Two factors explain that $O_3$ is not formed near the source of precursors. First, given the urban environment of the basin, $NO_x$ are emitted in large quantities so that the VOC/$NO_x$ ratio is adverse to neutral, and $O_3$ is formed in small amounts (see the related discussion in Sect. 3.2.2). But $NO_x$ have a lifetime much shorter than most VOC (e.g. Monod et al., 2001; Laughner and Cohen, 2019), and when the polluted air parcels are lifted up, the ratio becomes more balanced as $NO_x$ is consumed closer

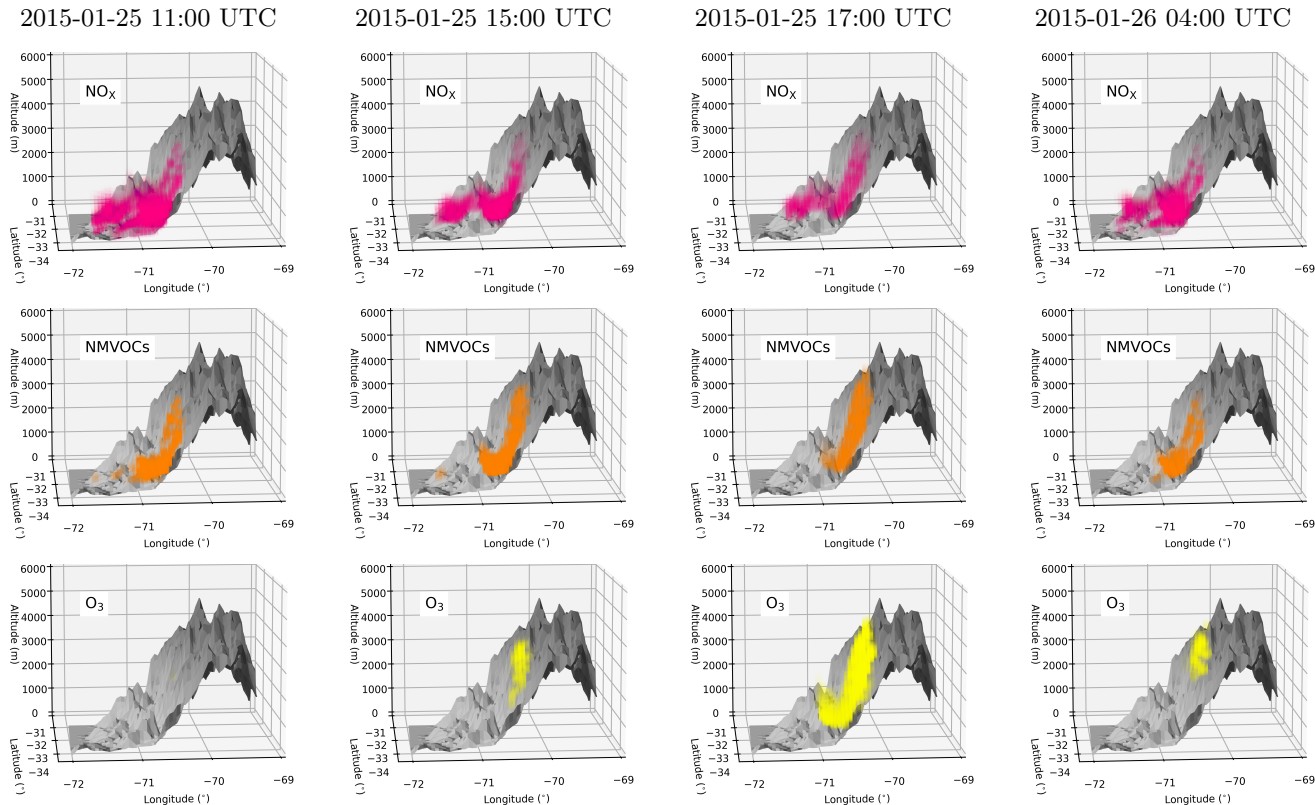

**Figure 10.** Cycle of $NO_x$ (top), NMVOC (middle) and $O_3$ (bottom) venting in the vicinity of Santiago for a typical day in January 2015. Hourly mixing ratios in excess of 2 ppb, 20 ppb and 60 ppb are shown, respectively. Each panel shows a 3-dimensional view from a longitudinal perspective. Gray surfaces represent the terrain topography. Santiago is UTC-3 for this period.

to the ground. Due to these asymmetric lifetimes and hence export, at some point on the vertical the ratio becomes favorable
(Fig. A5c) and $O_3$ is formed in large quantities. The second factor comes from the increase of photolysis rates with altitude. Several hundred meters above the ground, near or above the mixing layer, photolysis rates of $NO_2$ are much faster than at ground level (e.g. Pfister et al., 2000) hence favoring the formation of $O_3$, all other things being equal. In our simulation, on average, the photolysis of $NO_2$ is 20% faster 1000 m above ground than at the surface. Also, Figure 9d shows that at the point where the $O_3$ plume is denser, winds are weak to null on average, so that precursors stagnate, again allowing for more $O_3$
creation.

At 14:00 LT (17:00 UTC), the vertical export of precursors is even more important due to the maximum development of the deep mixing layer and the full intensity of upslope winds, with two main consequences. First, $NO_x$ levels in Santiago decrease compared to VOC due to vertical export earlier in the day, leading to a more favorable ratio and $O_3$ starts forming in the urban basin near the surface. Thus, ventilation of precursors possibly increases $O_3$ surface concentration in Santiago. Second,
the $O_3$ plume in the lower FT intensifies and extends up to an altitude of 4 km. Finally, at 01:00 LT (04:00 UTC) the next

day, the return of a shallower mixing layer and the accumulation of evening traffic emissions make precursors concentrations larger near the ground again. Sun being down, $O_3$ stops forming, but given the large amounts created during the day, a residual plume remains around 1 km above ground. In parallel, winds along the transect being mostly southerlies, the $O_3$ plume ends up approximately 50 km north of the Santiago basin.

## 4  Discussion

Figure 2 revealed moderate biases in the modeled concentrations of $PM_{2.5}$ and $O_3$ compared to downtown Santiago observations. These discrepancies can stem from (i) the relatively coarse resolution of the simulation compared to the heterogeneity of pollution at the scale of Santiago city, (ii) the static nature of the emissions inventory as of 2010 while air pollution follows a decreasing trend in Santiago hence accounting for the overestimation of primary emissions by the model, (iii) a slight negative bias in the representation of mixing layer height in the simulation contributing to over-concentrate particulate matter. In practice, a combination of the three is likely at play. Although the processes of transport evidenced in this study are not sensitive to such biases, quantitative conclusions can depend on them to some extent. In particular, if case (ii) dominates the discrepancy on $PM_{2.5}$, biases in emissions can be asymmetric between Santiago and other locations given that the trends since 2010 are probably not identical (the model in Rancagua is negatively biased for example). In that case, an overestimation of the relative contribution of Santiago emissions at these locations could be found. If the bias mostly comes from reason (i) or (iii), the quantification should be resilient: the Santiago source is considered at a larger-scale than downtown so that local discrepancies should compensate over the whole area (case (i)), and in case (iii) the bias should concern most locations, and does not have an influence on emissions, and therefore should not modify relative contributions. Regarding the biases on $O_3$ mixing ratio, case (i) is most likely to be the underlying bias, as levels are better reproduced in the eastern part of Santiago (Las Condes) thus pointing to localized disagreements. More generally, we cannot exclude possible shortcomings of the chemistry-transport model itself to explain the disagreement. Although CHIMERE has been systematically evaluated over Europe, it has not been extensively used over South America so far. In that case, identifying the causes and consequences is more complex.

Only one particular month per season, for one particular year, are analyzed here. Whether the results presented can be extrapolated with a climatological relevance is not straightforward. Climate variability modes such as ENSO, for instance, can lead to inter-annual changes in circulation and meteorology at the scale of central Chile. In particular, El Niño phases are related to above average rainfall in the region in winter (Montecinos and Aceituno, 2003), hence leading to more frequent scavenging of particulate matter, that may imply less regional transport. La Niña years do the opposite, with a dryer winter. In addition, ENSO affects the southeast Pacific anticyclone thus modulating wind speeds alongshore central Chile (Rahn and Garreaud, 2014). Weaker winds are observed in case of El Niño so that northward transport likely decreases as a consequence. Again, the opposite can be said for La Niña. ENSO also modulates the severity of the fire season in summertime in the region (Urrutia-Jalabert et al., 2018). However, as pointed out in Section 1, wildfires are not taken into account in our simulation design so that our findings are resilient to this variability. Speculating on the impacts regarding our results of the ENSO-related synoptic-scale variability of atmospheric circulation is even more complex in the context of the last decade, as the ENSO teleconnection in

central Chile is weak for that period (Garreaud et al., 2020). However, although our quantitative conclusions may not exactly hold for other years, the underlying processes described remain valid, particularly when it comes to mountain-valley circulation which is radiatively driven. Variations in primary pollutants and precursors emissions, wintertime precipitation and cloud cover may affect our results, but we do not expect new processes to take place or evidenced processes to stop, nor magnitudes to change entirely. Indeed, the year 2015 was chosen because it recorded no particular extreme pollution event, and corresponds to a neutral ENSO phase for the Chilean climate. In addition, primary emissions in the model are not weather-dependent, and the inventory is static as of 2010, meaning the fluxes of primary anthropogenic emissions would be equal for any simulated year. Therefore, the picture provided by this study is statistically representative of average summer and winter months, despite not being directly extendable.

Section 3.3 evidences the asymmetric vertical ventilation of pollutants such as $NO_x$ and VOC and its role in the formation of an $O_3$ persistent plume. Given the quite unique combination of very high emission rates and close proximity of steep elevated orography featured by the Santiago basin, it is unclear, based on the literature, whether this venting is expected to improve or worsen $O_3$ pollution at the surface level in the city. To some extent, this asymmetric venting of precursors is similar to the sensitivity analysis performed in Section 3.2 leading to a change in precursors ratios, except for the magnitude of the variation in pollutants emissions. Thus, one could extrapolate and expect an increase (decrease, respectively) of $O_3$ mixing ratio in the western (eastern, respectively) districts due to this daily-occurring export of pollutants, compared to a situation where air masses would stagnate. However, there are too many non-linear processes involved to be affirmative on this point.

In addition to being a pollutant, tropospheric $O_3$ is a strong greenhouse gas, estimated to contribute between 0.2 to 0.6 $W\,m^{-2}$ to present climate global radiative forcing (Myhre et al., 2013). The process described above may then imply radiative effects, either directly through greenhouse effect, indirectly through its impact on moisture, clouds and atmospheric circulation, or through its interaction with the cycles of other greenhouse gases (Mohnen et al., 1993). Estimating this impact is however not in the scope of the present work. Nevertheless, it shows that although the venting of pollutants can be beneficial from the urban air pollution perspective, longer-term effects on climate are a corollary worth investigating.

Despite our good confidence in the model, it is not possible to strengthen conclusiveness with observations on the newly evidenced $O_3$ bubble we find as there are no local measurements of $O_3$ profiles for our study period. Besides, tropospheric ozone column products from satellite data are usually not fit for analysis in mountainous regions (e.g. Kar et al., 2010). However, Seguel et al. (2013) conducted ozone profile measurements in the Santiago Metropolitan Region in summer 2011 that match our results quite well. First, they find a similar 35 ppb free troposphere background $O_3$ mixing ratio. Second, they evidence several occurrences of deep residual layers as intense as 100 ppb of $O_3$ slightly north of Santiago (at the location of La Colina, between Santiago and Los Andes) in early afternoon, measured between 1.5 km and 2.5 km above ground. Such secondary layers higher up are consistent with our findings. If a vertical profile is taken north of Santiago in Fig 9d, a bell shaped profile is recovered, of maximum intensity near 2 km above ground, much like Figure 9 in Seguel et al. (2013). Thus, our simulation results agree very well with the measurements conducted in Seguel et al. (2013), despite being for a different year, hence strengthening our conclusions on the existence of the newly evidenced $O_3$ bubble and its seasonal persistence. Similarly to our findings, Seguel et al. (2013) also concluded that the residual layer is coming from pollutants venting from

Santiago. However their measurement-based approach did not allow to evidence (i) the exact formation mechanism of this bubble, (ii) its persistent character even through nighttime (measurements presented are only for daytime), (iii) its horizontal extent (measurements are discrete in space). Our modeling approach based on sensitivity analysis confirms the primary role of Santiago emissions and gives a clearer and continuous 3-dimensional picture of the phenomenon, while agreeing with the findings of these previous measurements.

We estimate that 14% of $PM_{2.5}$ in Santiago come from long-range transport in wintertime. Although summertime $PM_{2.5}$ is not discussed within the framework of this paper, it is available in the simulations, and we find its transported contribution to be greater, at 22%, for that period. As a result, we expect the average contribution of long-range transport to $PM_{2.5}$ in Santiago for a whole year to be somewhere between these two numbers, around 18%. It is worth noting that for the year 2019, the exceedance of the $PM_{2.5}$ Chilean standard was 18% for Santiago according to data from the SINCA network. Although for our study year 2015 this exceedance was higher (close to 50%), our findings suggest that if air quality improvement policies are conducted in all urban areas across central Chile, Santiago might be able to meet the national standards more easily due to the large contribution of transported $PM_{2.5}$.

Summertime $PM_{2.5}$ and wintertime $O_3$ were not studied here given their comparatively lower relevance. A short discussion can be provided however. The simulations show that the regional plume of $PM_{2.5}$ in summertime has a similar extent to the one of $O_3$ in summertime, i.e. an area of influence lesser than in wintertime, mostly near the surface. $O_3$ is barely produced in wintertime due to a zenith angle of the sun closer to the horizon and shorter duration of days, resulting in less radiative power and barely active photolysis, so that the question of the export of its precursors at that season is not relevant. The model accordingly yields generally inconsequential mixing ratios in the domain.

## 5 Conclusions

Based on chemistry-transport modeling with WRF-CHIMERE, the present work investigates the transport of atmospheric pollutants in central Chile for one winter month and one summer month. Our findings show that emissions from Santiago city greatly affect atmospheric composition in its vicinity and farther, with a contribution of a few $\mu g\,m^{-3}$ to $PM_{2.5}$ in wintertime corresponding to 5% to 10% of surface concentrations as far as 500 km to the north and 500 km to south. This transport is mostly driven by surface winds within the boundary layer above land, and takes place in the free troposphere over the ocean hence explaining its long northward range. The spatial extent of the effect on surface concentrations of $O_3$ precursors emitted in Santiago in summertime is lesser. Nevertheless, daily peaks of $O_3$ in the direct vicinity of Santiago are reduced by up to 50 ppb on average when emissions from the city are eliminated. Conversely, the contribution of long-range transport of $PM_{2.5}$ in wintertime is responsible for $5\,\mu g\,m^{-3}$ to $10\,\mu g\,m^{-3}$ on average in downtown Santiago, corresponding to around 14% of the baseline concentration. While the transport of $O_3$ precursors to Santiago in summertime is not significant, if emissions of precursors were to be decreased in the city, its western districts would see their $O_3$ mixing ratios increase on average, despite daily peaks largely dropping, while the eastern area would improve for every quantile. This phenomenon is linked to heterogeneous emissions within the city, which make for currently higher than background levels in the east, and lower than

background in the west. When all emissions are cut, the whole area is brought to background, hence the respective variations. The vertical export of precursors above Santiago in summertime, in relation with unperturbed mountain-valley circulation and venting, creates a persistent $O_3$ bubble of more than 50 ppb on average, around 1000 m above the ground, slightly north of
510 the city. Daytime upslope winds, the heterogeneous lifetimes of precursors, and increasing vertical profiles of photolysis rate account for this formation, which impact should be looked at in greater detail in terms of surface air quality improvement (or worsening), and photo-chemical and greenhouse effects.

*Code availability.* The CHIMERE model used can be found at http://www.lmd.polytechnique.fr/chimere/CW-download.php (last access 1 December 2020). The WRF model used can be found at http://www2.mmm.ucar.edu/wrf/users/download/get_source.html (last access 1
December 2020).

*Data availability.* Surface observation data used in this study are available at https://sinca.mma.gob.cl/index.php/region/index/id/M (last access 1 December 2020). HTAP raw emission inventory can be downloaded at http://edgar.jrc.ec.europa.eu/htap_v2/ (last access 1 December 2020). Other data can be made available from the corresponding author upon reasonable request.

*Author contributions.* LM and SM supervised the chemistry transport simulations and analysis of the results. NH participated in the critical
analysis of the results. RL performed the conceptualization, data analysis and model simulations, and coordinated the writing of the paper with LM, SM and NH.

*Competing interests.* The authors declare that they have no conflict of interest.

*Acknowledgements.* The chemistry-transport simulations used in this work were performed using the high-performance computing resources of TGCC (Très Grand Centre de Calcul du CEA) under the allocation GEN10274 provided by GENCI (Grand Équipement National de Calcul
Intensif). The authors acknowledge the Editor and the two anonymous Reviewers for their valuable comments and feedback.

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

| WRF configuration | | CHIMERE configuration | |
| --- | --- | --- | --- |
| Horizontal resolution | 5km | Horizontal resolution | 5km |
| Vertical levels | 60 | Vertical levels | 30 |
| Microphysics | WSM3 | Chemistry | MELCHIOR |
| Boundary and surface layer | MYNN | Gas/Aerosol Partition | ISORROPIA |
| Land surface | Noah LSM | Horizontal Advection | Van Leer |
| Cumulus parameterization | Grell G3 | Vertical Advection | Upwind |
| Longwave radiation | CAM | Boundary Conditions | LMDz-INCA + GOCART |
| Shortwave radiation | Dudhia | | |

**Table A1.** WRF and CHIMERE configurations.

| | Wintertime | | | | | | | Summertime | | | | | | |
| --- | --- | --- | --- | --- | --- | --- | --- | --- | --- | --- | --- | --- | --- | --- |
| Station | Independencia | | | | Rancagua | | | Independencia | | | | Rancagua | | |
| | MB | NRMSE | R | | MB | NRMSE | R | MB | NRMSE | R | | MB | NRMSE | R |
| T2 | -1.23 | 0.22 | 0.87 | : | -1.65 | 0.28 | 0.7 | 0.4 | 0.16 | 0.75 | : | -0.06 | 0.19 | 0.73 |
| RH | -0.22 | 0.39 | 0.85 | : | -0.17 | 0.35 | 0.85 | -0.12 | 0.3 | 0.83 | : | -0.12 | 0.4 | 0.8 |
| U10 | -0.11 | 0.36 | 0.52 | : | -0.01 | 0.2 | 0.77 | 0.31 | 0.84 | -0.06 | : | 0.32 | 0.75 | 0.09 |
| V10 | -0.03 | 0.27 | 0.82 | : | 0.28 | 0.12 | 0.87 | 1.09 | 2.11 | 0.65 | : | 0.03 | 0.33 | 0.4 |
| Station | Los Andes | | | | Viña del Mar | | | Andacollo | | | | Viña del Mar | | |
| | MB | NRMSE | R | | MB | NRMSE | R | MB | NRMSE | R | | MB | NRMSE | R |
| T2 | -3.33 | 0.38 | 0.89 | : | 0.31 | 0.3 | 0.25 | 0.67 | 0.17 | 0.8 | : | 0.64 | 0.42 | 0.18 |
| RH | -0.17 | 0.3 | 0.91 | : | -0.17 | 0.74 | 0.47 | 0.0 | 0.13 | 0.81 | : | -0.06 | 0.57 | 0.51 |
| U10 | 0.29 | 0.39 | 0.77 | : | -0.12 | 1.35 | 0.29 | 0.85 | 1.36 | -0.08 | : | 0.83 | 2.68 | 0.31 |
| V10 | 0.74 | 0.65 | 0.58 | : | 2.12 | 1.54 | 0.61 | 2.14 | 1.14 | 0.25 | : | 0.96 | 1.57 | 0.64 |

**Table A2.** Simulation scores for daily average low-level meteorology for wintertime and summertime 2015. T2 is the 2 m air temperature (°C), RH the surface relative humidity, U10 10 m zonal wind and V10 10 m meridional wind speed (m s$^{-1}$). MB is the mean bias, NRMSE the normalized root mean square error and R the Pearson correlation coefficient.

| Day: | | 21 July | | | | 23 July | |
|---|---|---|---|---|---|---|---|
| | MB | NRMSE | R | | MB | NRMSE | R |
| TEMP | -0.3 | 0.03 | 1.0 | : | 1.13 | 0.07 | 1.0 |
| RH | -0.17 | 0.32 | 0.79 | : | -0.04 | 0.25 | 0.48 |
| U | 1.29 | 0.16 | 0.89 | : | -1.65 | 0.35 | 0.56 |
| V | -0.46 | 0.2 | 0.82 | : | 3.52 | 0.31 | 0.83 |
| Day: | | 24 July | | | | 25 July | |
| | MB | NRMSE | R | | MB | NRMSE | R |
| TEMP | 2.36 | 0.08 | 1.0 | : | 1.15 | 0.06 | 1.0 |
| RH | -0.03 | 0.53 | 0.01 | : | -0.04 | 0.17 | 0.89 |
| U | -0.52 | 0.14 | 0.86 | : | -0.51 | 0.16 | 0.86 |
| V | 0.62 | 0.13 | 0.92 | : | 3.35 | 0.29 | 0.86 |

**Table A3.** Simulation scores for meteorological vertical profiles for four days in July 2015 at noon local time at Quinta Normal station in Santiago. The same notations as in Table A2 apply.

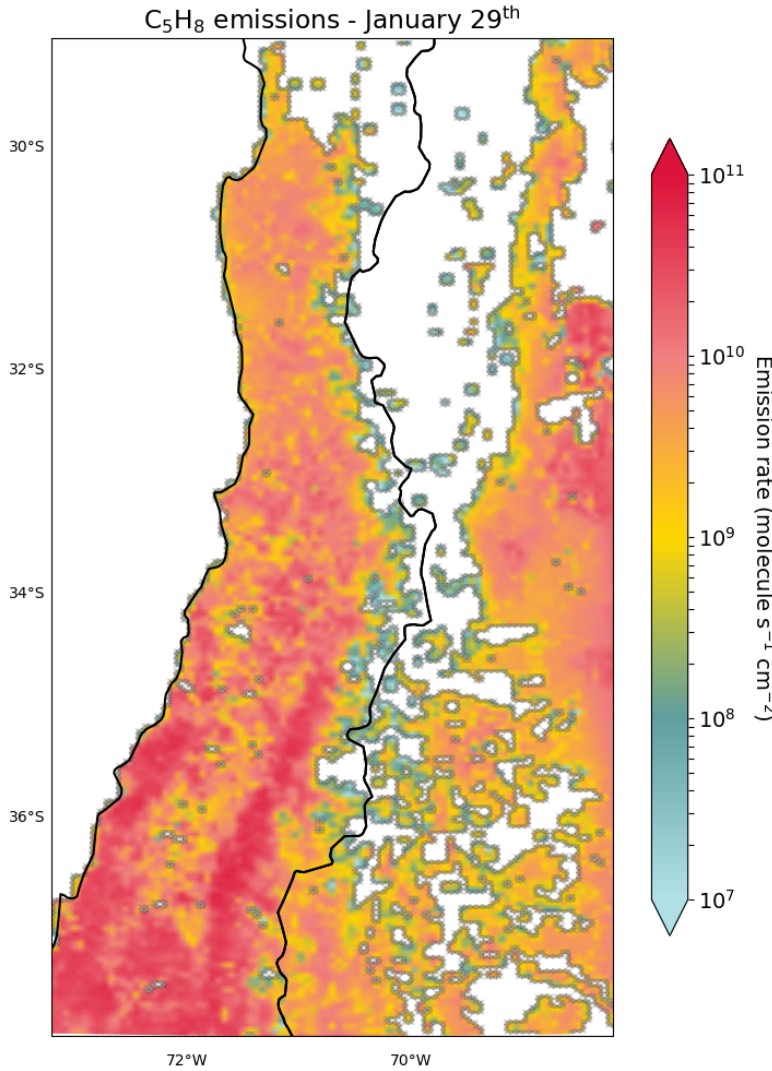

**Figure A1.** Isoprene ($C_5H_8$) average emission rate from biogenic sources for January 29[th] as computed in CHIMERE using the MEGAN model.

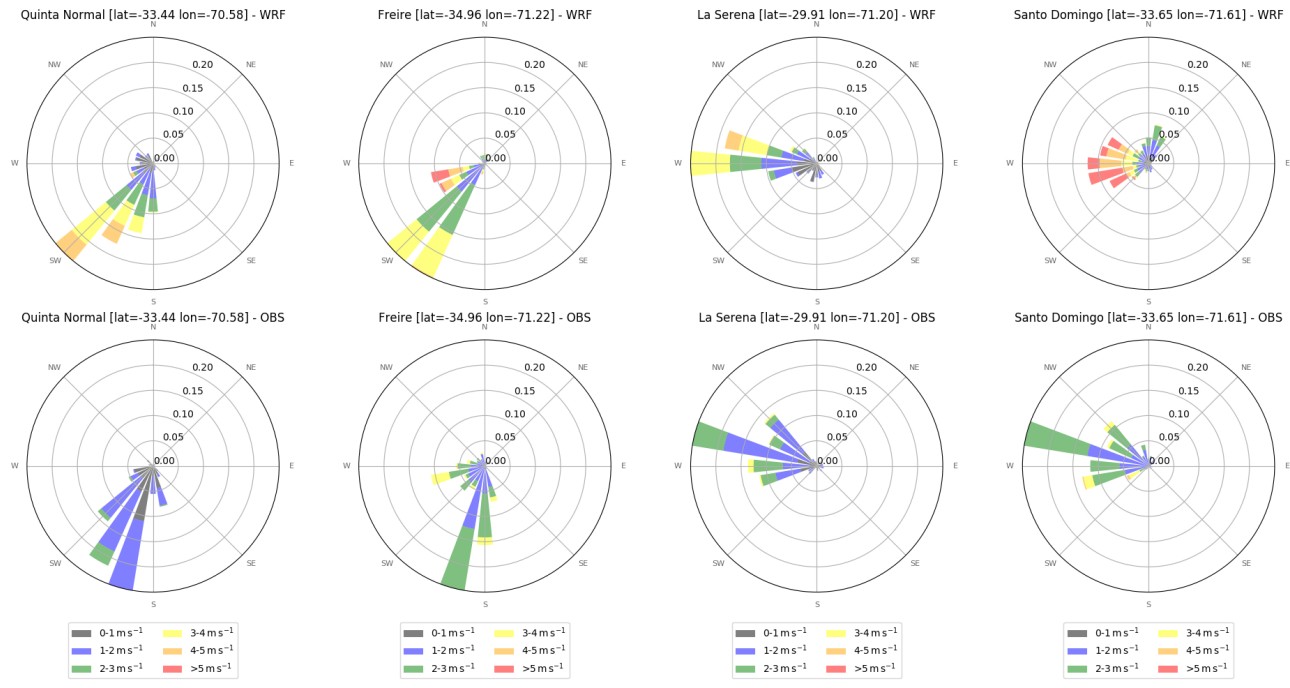

**Figure A2.** 10 m wind distribution in the simulation (top) and observations (bottom) at four locations in central Chile for summertime 2015.

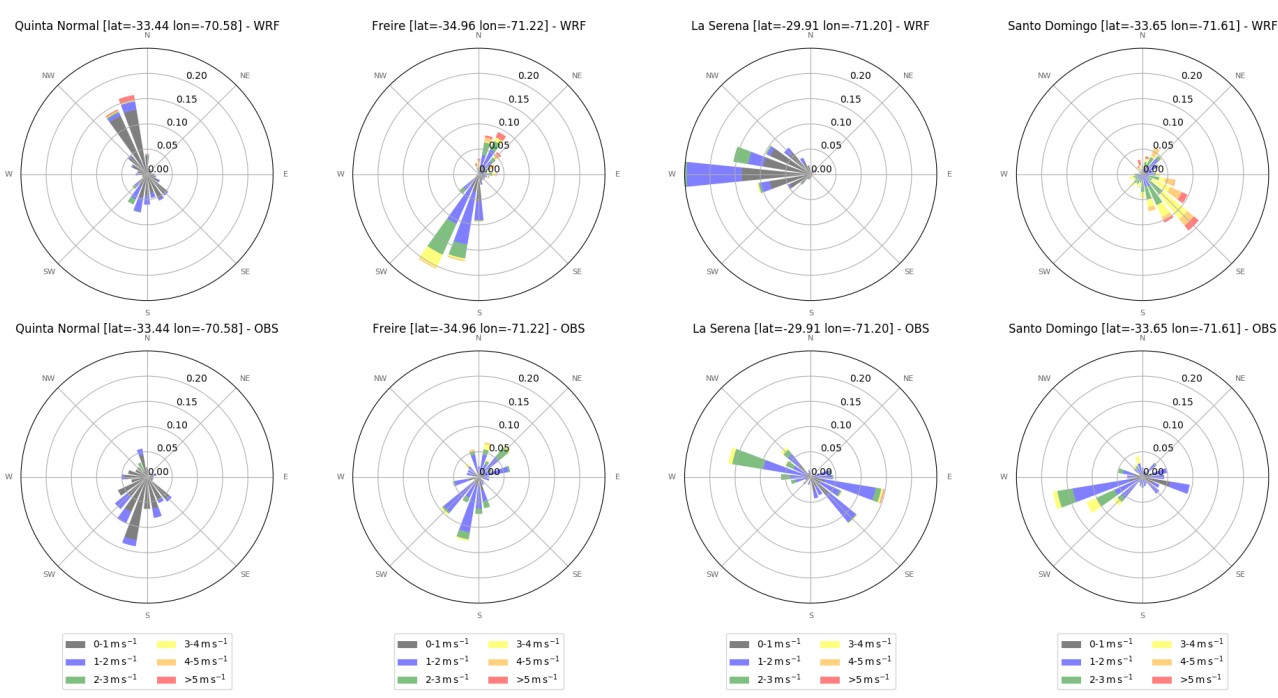

**Figure A3.** Same as Figure A2 for wintertime 2015.

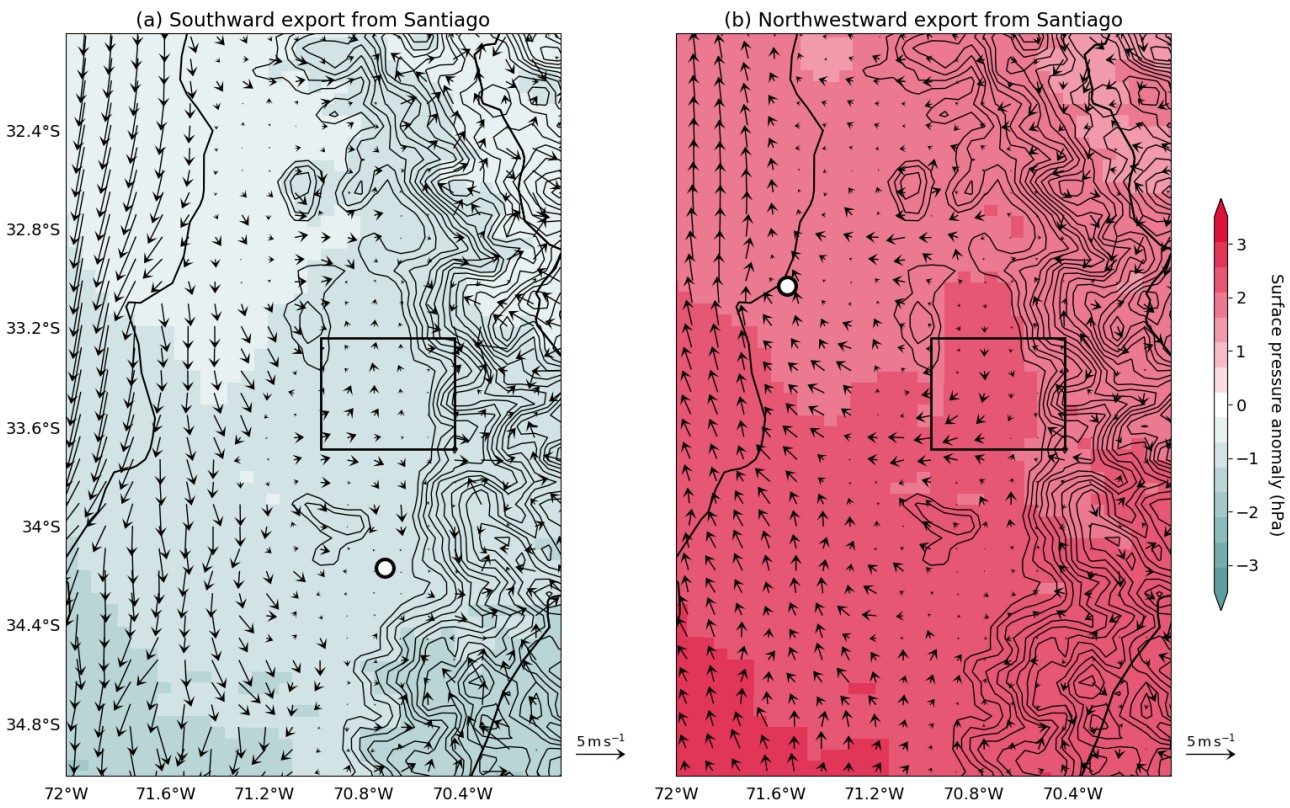

**Figure A4.** (a) Composite of surface wind (arrows) and pressure (colormap) anomalies with respect to the wintertime average, for hours of southward transport episodes, defined as hours when the concentration of $PM_{2.5}$ from Santiago (black rectangle) in Rancagua (white dot) is greater or equal to its $90^{th}$ percentile for the period. Contours show terrain elevation in excess of 1000 m every 250 m. (b) same as (a) but during hours of northwestward transport, defined in the same manner but with respect to concentrations in Viña del Mar (white dot).

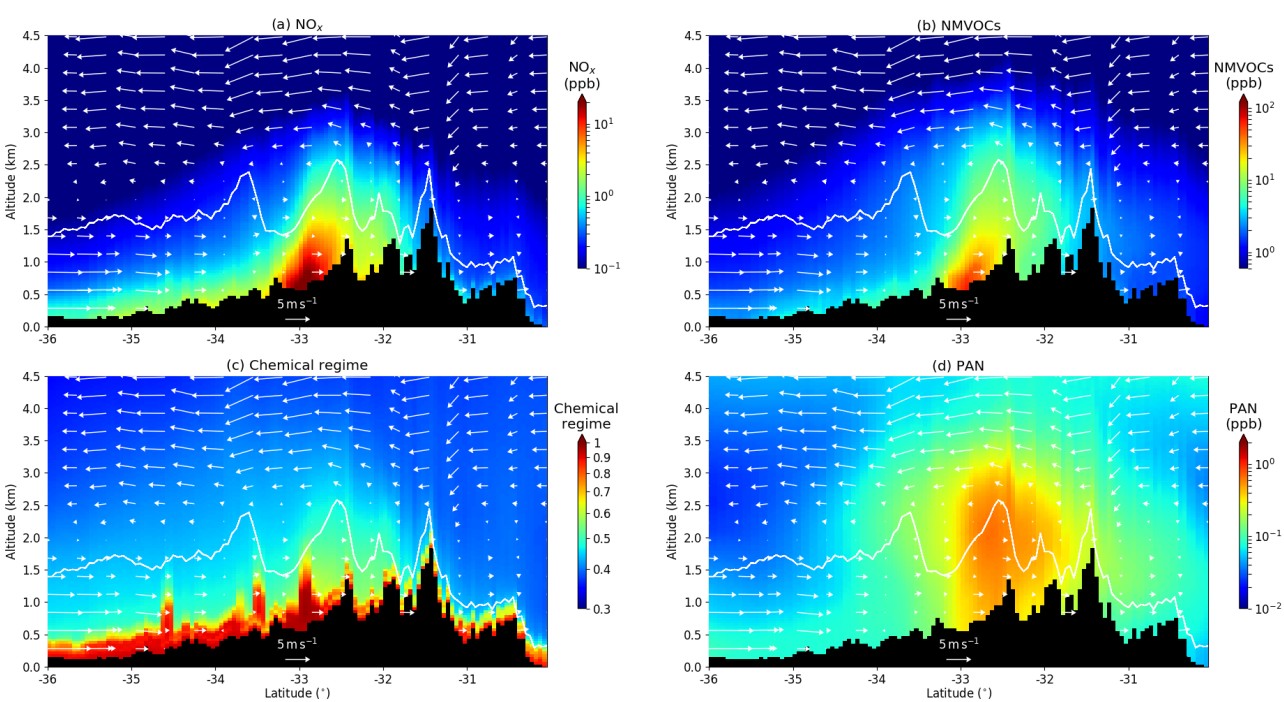

**Figure A5.** (a) Simulated NO$_x$ concentrations (colormap), wind (arrows), afternoon boundary layer (white solid line) and terrain elevation (black area), along the summertime transect considered in Figure 9a. Average for summertime 2015. (b) same as (a) for NMVOC, (c) same as (a) for chemical regime, (d) same as (a) for Peroxyacetyl nitrate (PAN).