# Peer review of "Seasonal variation of atmospheric pollutants transport in central Chile: dynamics and consequences"

_Atmospheric Chemistry and Physics, 2020_

## Author Comment (AC1)

**Answers to reviewers for ACP-2020-1249**

March 19, 2021

**Seasonal variation of atmospheric pollutants transport in central Chile: dynamics and consequences**

Rémy Lapere et al.

Dear Editor and Reviewers,

We acknowledge the Editor and Reviewers for the time spent to evaluate our work and for their valuable comments. We made the proposed changes in the revised manuscript. Please note that answers are in blue and after each Reviewer's remark, and sentences added/adjusted in the manuscript are quoted in *[italic font between brackets]*. All comments were addressed and are detailed in this letter. Summarizing our answers:

- 1. Discussions on the implications of inter-annual climate variability and model biases for the generalization and robustness of our results were added to the manuscript.
- 2. Additional details regarding the modeling methodology and the locations considered throughout the study were incorporated.
- 3. The organization of the manuscript has been improved by separating more clearly the elements of discussion (gathered into the Discussion section) from the Results section.
- 4. Rewording was performed where needed to enhance readability and proposed references were included for a better illustration of the context of this research.

NB: lines and figures numbers are to be understood in reference to the first submitted manuscript, so as to be consistent with the Reviewers' comments.

Best regards, Rémy Lapere March 19, 2021

**Anonymous Referee #1 https://doi.org/10.5194/acp-2020-1249-RC2**

**Major comments:**

1. The study is focused on the average transport patterns and the average contribution of emissions from Santiago to regional pollutant levels, but there is considerable variability in these numbers, which is not discussed. For example, the average contribution of Santiago emissions to wintertime PM2.5 in Rancagua is 1-2 µg/m3 but the maximum is as high as 20 µg/m3. The authors could identify what kind of transport patterns cause such high contributions downwind of Santiago, and what kind of transport patterns bring the most pollution from the surroundings into Santiago.

**Answer:**

Thank you for suggesting this interesting piece of investigation. We propose to include an analysis of wintertime largest PM import/export events with the following paragraph, in section 3.3, after line 335, along with Figure 1 below (Figure A3 in the new version of the manuscript):

In this wintertime averaged picture, the patterns underlying the largest transport events are not obvious. In order to identify these advection patterns, four clusters are designed corresponding to episodes of transport from Santiago to the south (Rancagua), from Santiago to the northwest (Viña del Mar), into Santiago from the south, and into Santiago from the northwest. In the case of transport into Santiago, passive aerosol tracers emitted in the model at the locations corresponding to Rancagua and Viña del Mar are used to discriminate the main direction of origin. The composite of these episodes is defined as the hours when the  $PM_{2.5}$  (or tracer) contribution to concentration is greater or equal to its 90th percentile over the studied period. The meteorological conditions during these particular hours are then compared to the average for the whole period, in order to compute anomalies in the surface wind and pressure fields. Figure A3 shows that southward transport events are associated with lower than average surface pressure by 1 to 3hPa and large northerly anomalies over most of the wind field, and in particular in the corridor between Santiago and Rancaqua. Conversely, northwestward export is associated with positive anomalies in surface pressure and southerly anomaly in surface winds combined with easterly anomaly in the corridor between Santiago and Viña del Mar. The two aforementioned corridors are evidenced by the topographic contours in Figure A3. These variations are related to the dynamics of the southeast Pacific high, located near  $(35^{\circ}S, 110^{\circ}W)$ : a weakening or southward/westward displacement of the anticyclone lead to the anomalies observed in Figure A3a, while an opposite displacement leads to a situation similar to that of Figure A3b. Consistently, transport events into Santiago are related to symmetrical patterns (not shown here), with transport from the northwest featuring similar anomalies as in Figure A3a and transport from the south originating in the same anomalies as in Figure A3b.]

2. The simulated ozone and PM2.5 concentrations showed significant disagreement with observations for the Santiago station (Fig 2; 22 ppb for ozone and 16 µg/m3 for PM2.5). The authors could discuss the possible causes of this disagreement and, more importantly, implications for the subsequent results about the contribution of Santiago emissions to regional pollutants levels.

**Answer:**

We propose to include the following paragraph discussing the causes and implications of these discrepancies, in section 4 line 395:

---

## Author Comment (AC2)

**Answers to reviewers for ACP-2020-1249**

March 19, 2021

**Seasonal variation of atmospheric pollutants transport in central Chile: dynamics and consequences**

Rémy Lapere et al.

Dear Editor and Reviewers,

We acknowledge the Editor and Reviewers for the time spent to evaluate our work and for their valuable comments. We made the proposed changes in the revised manuscript. Please note that answers are in blue and after each Reviewer's remark, and sentences added/adjusted in the manuscript are quoted in *[italic font between brackets]*. All comments were addressed and are detailed in this letter. Summarizing our answers:

1. Discussions on the implications of inter-annual climate variability and model biases for the generalization and robustness of our results were added to the manuscript.

2. Additional details regarding the modeling methodology and the locations considered throughout the study were incorporated.

3. The organization of the manuscript has been improved by separating more clearly the elements of discussion (gathered into the Discussion section) from the Results section.

4. Rewording was performed where needed to enhance readability and proposed references were included for a better illustration of the context of this research.

NB: lines and figures numbers are to be understood in reference to the first submitted manuscript, so as to be consistent with the Reviewers' comments.

Best regards,
Rémy Lapere
March 19, 2021

**Anonymous Referee #1**
**https://doi.org/10.5194/acp-2020-1249-RC2**

**Major comments:**

1. The study is focused on the average transport patterns and the average contribution of emissions from Santiago to regional pollutant levels, but there is considerable variability in these numbers, which is not discussed. For example, the average contribution of Santiago emissions to wintertime PM2.5 in Rancagua is 1-2 µg/m3 but the maximum is as high as 20 µg/m3. The authors could identify what kind of transport patterns cause such high contributions downwind of Santiago, and what kind of transport patterns bring the most pollution from the surroundings into Santiago.

   *Answer:*
   Thank you for suggesting this interesting piece of investigation. We propose to include an analysis of wintertime largest PM import/export events with the following paragraph, in section 3.3, after line 335, along with Figure 1 below (Figure A3 in the new version of the manuscript):

   *[In this wintertime averaged picture, the patterns underlying the largest transport events are not obvious. In order to identify these advection patterns, four clusters are designed corresponding to episodes of transport from Santiago to the south (Rancagua), from Santiago to the northwest (Viña del Mar), into Santiago from the south, and into Santiago from the northwest. In the case of transport into Santiago, passive aerosol tracers emitted in the model at the locations corresponding to Rancagua and Viña del Mar are used to discriminate the main direction of origin. The composite of these episodes is defined as the hours when the $PM_{2.5}$ (or tracer) contribution to concentration is greater or equal to its $90^{th}$ percentile over the studied period. The meteorological conditions during these particular hours are then compared to the average for the whole period, in order to compute anomalies in the surface wind and pressure fields. Figure A3 shows that southward transport events are associated with lower than average surface pressure by 1 to 3 hPa and large northerly anomalies over most of the wind field, and in particular in the corridor between Santiago and Rancagua. Conversely, northwestward export is associated with positive anomalies in surface pressure and southerly anomaly in surface winds combined with easterly anomaly in the corridor between Santiago and Viña del Mar. The two aforementioned corridors are evidenced by the topographic contours in Figure A3. These variations are related to the dynamics of the southeast Pacific high, located near (35° S, 110° W): a weakening or southward/westward displacement of the anticyclone lead to the anomalies observed in Figure A3a, while an opposite displacement leads to a situation similar to that of Figure A3b. Consistently, transport events into Santiago are related to symmetrical patterns (not shown here), with transport from the northwest featuring similar anomalies as in Figure A3a and transport from the south originating in the same anomalies as in Figure A3b.]*

2. The simulated ozone and PM2.5 concentrations showed significant disagreement with observations for the Santiago station (Fig 2; 22 ppb for ozone and 16 µg/m3 for PM2.5). The authors could discuss the possible causes of this disagreement and, more importantly, implications for the subsequent results about the contribution of Santiago emissions to regional pollutants levels.

   *Answer:*

   We propose to include the following paragraph discussing the causes and implications of these discrepancies, in section 4 line 395:

[Figure]

Figure 1: (a) Composite of surface wind (arrows) and pressure (colormap) anomalies with respect to the wintertime average, for hours of southward transport episodes, defined as hours when the concentration of PM$_{2.5}$ from Santiago (black rectangle) in Rancagua (white dot) is greater or equal to its 90% percentile for the period. Contours show terrain elevation in excess of 1000 m every 250 m. (b) same as (a) but during hours of northwestward transport, defined in the same manner but with respect to concentrations in Viña del Mar (white dot).

*[Figure 2 revealed moderate biases in the modeled concentrations of PM$_{2.5}$ and O$_3$ compared to downtown Santiago observations. These discrepancies can stem from (i) the relatively coarse resolution of the simulation compared to the heterogeneity of pollution at the scale of Santiago city, (ii) the static nature of the emissions inventory as of 2010 while air pollution follows a decreasing trend in Santiago hence accounting for the overestimation of primary emissions by the model, (iii) a slight negative bias in the representation of mixing layer height in the simulation contributing to over-concentrate particulate matter. In practice, a combination of the three is likely at play. Although the processes of transport evidenced in this study are not sensitive to such biases, quantitative conclusions can depend on them to some extent. In particular, if case (ii) dominates the discrepancy on PM$_{2.5}$, biases in emissions can be asymmetric between Santiago and other locations given that the trends since 2010 are probably not identical (the model in Rancagua is negatively biased for example). In that case, an overestimation of the relative contribution of Santiago emissions at these locations could be found. If the bias mostly comes from reason (i) or (iii), the quantification should be resilient: the Santiago source is considered at a larger-scale than downtown so that local discrepancies should compensate over the whole area (case (i)), and in case (iii) the bias should concern most locations, and does not have an influence on emissions, and therefore should not modify relative contributions. Regarding the biases on O$_3$ mixing ratio, case (i) is most likely to be the underlying bias, as levels are better reproduced in the eastern part of Santiago (Las Condes) thus pointing to localized disagreements. More generally, we cannot exclude possible shortcomings of the chemistry-transport model itself to explain the disagreement. Although CHIMERE has been systematically evaluated over Europe, it has not been extensively used over South America so far. In that case, identifying the causes and consequences is more complex.]*

3. The analysis is based on just 1 month of simulation in each season, which is a major limitation. The authors briefly discuss this point in section 4, but a more thorough discussion will be appreciated, particularly of the synoptic-scale circulation patterns in relation to ENSO. Analysis of the NCEP FNL data could help in this.

*Answer:*

We propose to modify the first paragraph of section 4 to include the additional elements of discussion required, based on literature. It now reads, from line 397:

*[... at the scale of central Chile. In particular, El Niño phases are related to above average rainfall in the region in winter [Montecinos and Aceituno, 2003], hence leading to more frequent scavenging of particulate matter, that may imply less regional transport. La Niña years do the opposite, with a dryer winter. In addition, ENSO affects the southeast Pacific anticyclone thus modulating wind speeds alongshore central Chile [Rahn and Garreaud, 2014]. Weaker winds are observed in case of El Niño so that northward transport likely decreases as a consequence. Again, the opposite can be said for La Niña. ENSO also modulates the severity of the fire season in summertime in the region [Urrutia-Jalabert et al., 2018]. However, as pointed out in Section 1, wildfires are not taken into account in our simulation design so that our findings are resilient to this variability. Speculating on the impacts regarding our results of the ENSO-related synoptic-scale variability of atmospheric circulation is even more complex in the context of the last decade, as the ENSO teleconnection in central Chile is weak for that period [Garreaud et al., 2020]. However, although our quantitative conclusions may not exactly hold for other years, the underlying processes described remain valid, particularly when it comes to mountain-valley circulation which is radiatively driven. Variations in primary pollutants and precursors emissions, wintertime precipitation and cloud cover may affect our results, but we do not expect new processes to take place or evidenced processes to stop, nor magnitudes to change entirely. Indeed, the year 2015 was chosen because it recorded no particular extreme pollution event, and corresponds to a neutral ENSO phase for the Chilean climate. In addition, primary emissions in the model are not weather-dependent, and the inventory is static as of 2010, meaning the fluxes of primary anthropogenic emissions would be equal for any simulated year. Therefore, the picture provided by this study is statistically representative of average summer and winter months, despite not being directly extendable.]*

**Minor and technical comments:**

4. Line 45 states that wildfire emissions are not considered. This is not clear. Are they completely ignored, even in the baseline simulation? That wouldn't make sense. The authors should clarify this.

*Answer:*

Wildfire emissions are indeed completely ignored in all simulations performed in this study. We argue that including this emission source is not relevant in the framework of this study for the following reasons:

- Wildfire emissions regard only the summertime simulation, and primarily affect PM concentration in the area (which is not studied here for this season). They can affect ozone levels in the locations we are interested in, but to a marginal extent.

- The inter-annual variability of fires intensity in central Chile is large, with very different intensities from one year to another, which would make the generalization of the results even more complex to consider.

We propose to clarify that fires are ignored by changing the end of the sentence line 45 by:

*[... out of the scope of the present work and therefore ignored throughout this study.]*

5. Section 2.1: The method for the calculation of biogenic VOC emissions needs to be specified. A map of biogenic VOC emissions (maybe isoprene) would be useful.

***Answer:***
Additional sentences have been added at the end of the first paragraph of Section 2.1 to specify the method, along with a figure in Appendix (A1 - also shown below) showing an emission map of isoprene given by the model. The aforementioned additional paragraph reads:

*[Biogenic emissions fluxes in CHIMERE are computed online using the MEGAN (Model of Emissions of Gases and Aerosols from Nature) model [Guenther et al., 2006]. Emission fluxes from the vegetation are based on air temperature, photosynthetic photon flux density and leaf area index. As an example, an emission map of isoprene ($C_5H_8$, a VOC involved in the formation of $O_3$ and secondary organic aerosols) as computed in CHIMERE for a given day in January 2015 is shown in Figure A1, illustrating the meriodional gradient of vegetation cover.].*

[Figure]

Figure 2: Isoprene ($C_5H_8$) average emission rate from biogenic sources for January $29^{th}$ as computed in CHIMERE using the MEGAN model.

6. Section 2.1: It is not obvious which species are included in PM2.5. Does it include primary as well as secondary species? Does it include aerosol water? This could be clarified.

*Answer:*

The following sentence has been added to the third paragraph of Section 2.1 to clarify this point:

*[PM$_{2.5}$ include all primary aerosol species (including dust and sea-salt), as well as secondary organic aerosols, but do not incorporate aerosol water.]*

7. Lines 135-147: This paragraph is confusing. I understand the point the authors are trying to make about the lack of data to evaluate the simulated NOx and VOC. The authors should consider rewording this paragraph and be direct about it.

*Answer:*

We propose the following rewording for this paragraph:

*[The lack of available measurements for NO$_x$ and VOC in central Chile hinders the simulation validation regarding these precursors. However, the HTAP inventory has proved reliable for large urban basins in Argentina and Brazil in terms of magnitude of VOC emissions [Puliafito et al., 2017; Dominutti et al., 2020], and more generally all across southern South America for NO$_x$ [Huneeus et al., 2020]. Consequently, we postulate that emission rates input in the model are appropriate, hence providing adequate chemical regimes when it comes to the simulation of O$_3$ concentrations. Besides, the known biases of HTAP on these pollutants are critical when it comes to more detailed approaches for policy making but for the purpose of the present work, having the proper total amount is sufficient as we apply our own downscaling methodology, do not discuss very high-resolution processes, and rely mostly on sensitivity analysis.]*

8. Fig. 6(b): The colorbar's maximum value is missing.

*Answer:*

Thank you for pointing that out, the figure has been adjusted accordingly.

**Anonymous Referee #3**
**https://doi.org/10.5194/acp-2020-1249-RC1**

**General comments**

This study is good, and it has a large amount of work. However, it is not suitable for been published in Atmospheric Chemistry and Physics. This investigation presents limited information for a short-studied period, and the novelty of its methodology is not clear. The manuscript does not have the quality for been published in ACP neither. It is not easy to read in almost all sections. For example, the methodology section has many results, and the results section has extensive discussions. Indeed, results, conclusion and perspectives are not clear neither. The document is also confusing describing the studied areas. It mentioned cities/towns that someone who does not know much about Santiago geography will not understand. I am not sure if the authors refer to the Santiago Metropolitan area and Central Chile as the same place. In my opinion, the authors developed a proper local scientific investigation for a specific period that needs to be improved to be published in this journal.

*Answer:*
We appreciate the comments formulated by the Reviewer, that helped improving the overall readability of the manuscript. Please find hereafter our answers to the above general comments:

- Short period of the study: this choice is extensively discussed in the manuscript, particularly with regard to the implications in terms of generalization of the results. We chose a year with no particular extreme pollution event and subject to a neutral ENSO phase. Given the methodology adopted, simulating a multi-year period would yield more variability in the results, but we argue that the average conclusions would remain similar, although a climatological extension is not straightforward. This discussion point has been expanded in the new version of the manuscript.

- Novelty of the methodology: although the sensitivity analysis to emissions with chemistry-transport modeling is not a novelty per se, applying this approach on the region of interest here is new, and thus helps evidence magnitudes of transported pollutants, impacted areas, and new phenomena (especially the persistent ozone plume in altitude) that are not found in the literature.

- Quality of the manuscript: based on both Reviewer's comments, the structure of the new version has improved. As for the geographical area of study, more details were provided as suggested.

**Specific comments**

**Abstract**

1. Line 3. This statement is not relevant to this air pollutant study.

   *Answer:*
   This statement speaks to the collateral consequences of air pollution in this region, hence justifying the need to study the transport patterns of pollutants. Although said consequences are indeed not directly investigated here, we believe it is an important element of context.

2. Line 11. I suggest to re-write '4o north and 4o south'.

   *Answer:*
   The following rewording has been made (both in abstract and conclusion) to avoid confusion regarding the use of latitude and longitude degrees:

*[... as far as 500 km to the north and 500 km to the south.]*

3. Line 15. What is a bubble formation mechanism? Although the word 'bubble' is mentioned in the manuscript, this mechanism is not explained.

    ***Answer:***
    The sentence was reworded as follows to enhance clarity:

    *[This work constitutes the first description of the mechanism underlying the latter phenomenon.]*

    **Introduction**

4. Line 27- Here, I suggest specifying the Region's name and cite the demographic information source. Chile has a Regional administrative division, and none of its regions has more than 8 million inhabitants. Do the authors refer to Central Chile area?

    ***Answer:***
    The suggested details have been included and precised. The beginning of this paragraph now reads:

    *[The central zone of Chile investigated in this study (referred to as 'central Chile' in the continuation) comprises the six administrative regions of Coquimbo, Valparaíso, Metropolitana de Santiago, O'Higgins, Maule and Ñuble. It is home to more than 12 million people [INE, 2018],...]*

5. Line 30 - I suggest moving the citation at the end of the sentence.

    ***Answer:***
    The citation has been moved accordingly.

6. Line 31. I suggest creating a new paragraph from "Tropospheric O3 . . . .

    ***Answer:***
    The paragraph has been split accordingly.

7. Line 37-38. The last sentence needs a reference.

    ***Answer:***
    The following reference has been added : [Ehhalt et al., 2001]

8. Line 46 to 48. I agree with this sentence. However, there is not a lack of information about air long-transport sources in Central Chile. Few studies show that Santiago has been reached by at least two long-transport sources: copper smelter and coastal air pollution. Some of these studies are: Barraza et al. 2017 (previously mentioned), Jorquera and Barraza 2012, Sci. Total Environ., 435-436, 418-429., Moreno et al., 2010. J. AirWaste Manage., 60, 1410-1421., Kavouras et al., 2001. J. Air Waste Manag., 51, 451-464, 2001., Gallardo et al., 2002. Atmos. Environ., 36, 3829-3841, 2002. Hedberg et al., 2005. Atmos. Environ., 39, 549-561.

    ***Answer:***
    We propose to account for this comment by modifying the sentence at line 46 as follows:

*[... Although specific studies revealed the importance of remote sources such as copper smelters [e.g. Gallardo et al., 2002; Hedberg et al., 2005; Moreno et al., 2010] and marine aerosols [e.g. Kavouras et al., 2001; Jorquera and Barraza, 2012; Barraza et al., 2017] in urban pollution in Chile, generally speaking the processes and patterns underlying pollutants transport are not well known, nor is the amount of advected contaminants.].*

**Data and methods**

9. Section 2.2. This section has several sentences with i) robustness of the results from the resolved model, ii) authors interpretation of used data, and iii) discussion. All this information is mixed with some input setting. I suggest making a difference between the methodology, outcomes, and data interpretation. Maybe adding a new sub-section "modelling results" would clarify the methodology used, results and interpretations.

   **Answer:**
   We understand the general idea of this comment, but we could not think of a better organization for this section. The "simulation validation" part may be considered as "modeling results" since it describes and discusses outputs of the model. In our opinion the results mentioned in this section only speak to the validity of the approach and do not describe yet the actual valuable modeling outcomes. The elements of discussion included here are marginal and designed to evidence that we can be confident in the results analyzed in the continuation. Thus, we believe that the structure as is (see paragraph by paragraph description below) provides necessary and sufficient information regarding the methodology, allowing to present the results within a confident framework. Although no changes were made here (except for shortening the discussion in paragraph 2.2.4 below), reorganizations were performed elsewhere in the manuscript according to the Reviewer's comments that should improve general readability.

   2.1 Modeling setup
   §1: models, domain, input data and settings involved
   §2: simulations design for the sensitivity analysis
   §3: justification of the relevance of the modeling setup/pollutants studied
   2.2 Simulation validation
   §1: presentation of the data used for validation
   §2: description and interpretation of the statistical scores for meteorology
   §3: description and interpretation of the statistical scores for surface pollutants
   §4: *a priori* validation of pollutants for which measurements are not available
   §5: conclusion paragraph summarizing the model validation

10. Line 92. I suggest you change the acronym to black carbon. It is the first-time using BC.

    **Answer:**
    The sentence line 29 has been adjusted to mention black carbon and its acronym directly in the Introduction and thus allow to use the acronym in line 92. It now reads:

    *[A side effect is the deposition of light-absorbing particulate matter, such as black carbon (BC), on the adjacent Andean snowpack...]*

11. Line 95. Here I suggest checking the name of the Region. Chile has a Regional administrative division, and none of its regions is named Region of Santiago. Do the authors refer to Santiago city or Metropolitan Region?

*Answer:*
*[Region of Santiago]* has been replaced with *[Santiago city]* in line 95.

12. Line 111. The sentence "the model is a little too dry" is relatively informal, and it does not sound like a technical interpretation or scientific information.

    *Answer:*
    This sentence line 111 has been replaced with *[The model shows a negative bias on surface relative humidity...]*.

13. Figure 1. I suggest changing BC to Black carbon (BC). It is the first-time using BC.

    *Answer:*
    See comment n°10

14. Figure 2. in my opinion, this is an interpretation of the results and not a methodology. I suggest mentioning this figure in the results section.

    *Answer:*
    This figure can arguably be described as results since it presents modeling outputs indeed. However, in line with comment n°9, we argue that these are not valuable model outputs per se, but rather a verification that the model behaves properly enough for the results described afterwards to be reliable. As a result, we deem it relevant in this section, even more so given the large number of figures already present in the results.

    **Results**

15. This section has a large amount of discussion. I suggest moving these sentences to the discussion section.

    *Answer:*
    Accordingly to the comment, we propose to move the following paragraphs from the results section to the discussion section:

    - lines 243 to 246: moved after line 434
    - lines 388 to 393: moved after line 409

16. Lines 152 to 165. Is this paragraph showing results of this study? This looks more like a description of the studied area, which will better suit the methodology section.

    *Answer:*
    This paragraph describes model outputs from the study, in terms of synoptic meteorology, that prelude, in a general manner, the more detailed results coming afterwards. In a way, it is a general description of the relevant meteorological parameters simulated by the model that will help to understand the findings related to atmospheric composition. General considerations and literature that are not specific to this study are used to provide a framework explaining the observations made in Figure 3. This paragraph does not, in our opinion, refer to methodological points but rather a specific introduction to the results immediately following. In order to improve readability, we propose to add the following introductory paragraph at line 151:

*[Hereafter are described well-known general meteorological features generated by the model, that provide a first clue regarding the advection of polluted air masses in the region, and constitute a reference frame accounting for the results described in the continuation.]*

17. Lines 195 to 197. This sentence needs a reference.

    **Answer:**
    Reference was added and the sentence line 195-197 now reads:

    *[... significant radiative effects when deposited [Rowe et al., 2019], especially given the large fraction of BC in PM$_{2.5}$ emitted in Santiago, around 15% according to the HTAP emissions inventory.]*

18. Line 197. I suggest changing "Andes cordillera" to "Andes mountains" (English) or "Cordillera de los Andes" (Spanish). It is better not to mix both languages.

    **Answer:**
    "Andes cordillera" is extensively used in the literature, including in titles of articles in high quality journals such as ACP. In this respect, when "cordillera" was initially used alone it has been replaced with "Andes" in order to comply with English requirements, but "Andes cordillera" occurrences were left unchanged.

19. Line 188. I understand that San Gabriel is a Mountain town and not a village. I suggest adding the altitude of both mentioned places.

    **Answer:**
    Altitudes of both sites have been added in line 188. We propose to use the neutral word "locality" to describe San Gabriel instead of "town", as there are only 2000 inhabitants and it is a district from a larger municipality.

20. Line 198. I suggest changing "ocean coast west" to "Central Chile coast at XXX km west from Santiago."

    **Answer:**
    The sentence line 198 has been modified accordingly and now reads:

    *[... second largest populated region of Chile, located on the coast of central Chile, approximately 100 km west of Santiago.]*

21. In my opinion, it is better to use the Andes instead of Cordillera (this is a Spanish word maybe not well known to an English audience)

    **Answer:**
    See comment n°18.

22. Line 280. If the authors add coordinates to Melipilla site, I suggest adding coordinates to the other places mentioned. As examples, Rancagua and San Fernando are not presenting coordinates (both previously mentioned in line 279).

    **Answer:**
    Coordinates were initially given for the Melipilla site as it does not appear on any map in the

manuscript, whereas all the other sites do, hence providing their location. That being said, coordinates of Rancagua and San Fernando have been added in the sentence line 279 for a more homogeneous paragraph.

23. Line 303. I suggest changing "center Santiago" to "Center of Santiago" or "Santiago Center."

    *Answer:*
    *[center Santiago]* has been changed to *[center of Santiago].*

24. Line 313 to 315. This sentence is confusing, in particular when the authors mentioned opposite ideas, such as" increase (decrease, respectively), and western (eastern, respectively).

    *Answer:*
    This sentence has been reworded and now reads:

    *[Therefore, this corresponds to an increase of $O_3$ in western Santiago and a decrease of $O_3$ in eastern Santiago, thus explaining...].*

25. In my opinion, it is better to use the Andes instead of Andes Cordillera (this is a Spanish word maybe not well known to an English audience)

    *Answer:*
    See comments n°18 and n°21.

    **Discussion**

26. This section looks like a continuation or a summary of the previous section. Result section has a lot of discussions; my suggestions are to combine both sections or remove the discussion from the results section.

    *Answer:*
    As suggested, we have moved discussion paragraphs from the results section to the discussion section (see reply to comment n°15). We have also included new discussion paragraphs following the comments from Referee #1 so that the discussion is now large and more independent from the results, and it makes sense to have it as a separate section.

27. Line 395 to 401. I can't entirely agree with this paragraph - the information presented in the current manuscript is not enough to support the next sentence "the result presented can be extrapolated...", mainly due to the short period studied.

    *Answer:*
    The complete sentence in the original manuscript is *[**Whether** the results presented can be extrapolated with a climatological relevance **is not straightforward**.],* meaning we think the results cannot be extrapolated in an easy manner. We believe the current formulation clearly states that point and already goes in the sense of the reviewer's opinion.

**Conclusion**

28. Line 449. This sentence sounds confusing: "East (the west, respectively) ".

    *Answer:*
    The sentence has been reworded and now reads:

    *[… which make for currently higher than background levels in the east, and lower than background in the west.]*

**References**

F. Barraza, F. Lambert, H. Jorquera, A. M. Villalobos, and L. Gallardo. Temporal evolution of main ambient $PM_{2.5}$ sources in Santiago, Chile, from 1998 to 2012. *Atmos. Chem. Phys.*, 17:10093–10107, 2017. doi: 10.5194/acp-17-10093-2017.

P. Dominutti, T. Nogueira, A. Fornaro, and A. Borbon. One decade of VOCs measurements in São Paulo megacity: Composition, variability, and emission evaluation in a biofuel usage context. *Sci. Total Environ.*, 738:139790, 2020. doi: 10.1016/j.scitotenv.2020.139790.

D. Ehhalt, M. Prather, F. Dentener, R. Derwent, E. Dlugokencky, E. Holland, I. Isaksen, J. Katima, V. Kirchhoff, P. Matson, P. Midgley, and M. Wang. *In: Climate Change 2001: The Physical Science Basis. Contribution of Working Group I to the Third Assessment Report of the Intergovernmental Panel on Climate Change*, chapter Atmospheric Chemistry and Greenhouse Gases. [Houghton, J.T., Y. Ding, D.J. Griggs, M. Noguer, P.J. van der Linden, X. Dai, K. Maskell, and C.A. Johnson (eds.)], Cambridge University Press, Cambridge, United Kingdom and New York, NY, USA, 2001.

L. Gallardo, G. Olivares, J. Langner, and B. Aarhus. Coastal lows and sulfur air pollution in Central Chile. *Atmos. Environ.*, 36:3829–3841, 2002. doi: 10.1016/S1352-2310(02)00285-6.

R. D. Garreaud, J. P. Boisier, R. Rondanelli, A. Montecinos, H. H. Sepúlveda, and D. Veloso-Aguila. The Central Chile Mega Drought (2010-2018): A climate dynamics perspective. *Int. J. Climatol.*, 40:421–439, 2020. doi: 10.1002/joc.6219.

A. Guenther, T. Karl, P. Harley, C. Wiedinmyer, P. I. Palmer, and C. Geron. Estimates of global terrestrial isoprene emissions using MEGAN (Model of Emissions of Gases and Aerosols from Nature). *Atmos. Chem. Phys.*, 6:3181–3210, 2006. doi: 10.5194/acp-6-3181-2006.

E. Hedberg, L. Gidhagen, and C. Johansson. Source contributions to PM10 andarsenic concentrations in Central Chile using positive matrix factorization. *Atmos. Environ.*, 39:549–561, 2005. doi: 10.1016/j.atmosenv.2004.11.001.

N. Huneeus, H. Denier van der Gon, P. Castesana, C. Menares, C. Granier, L. Granier, M. Alonso, M. F. Andrade, L. Dawidowski, L. Gallardo, D. Gomez, Z. Klimont, G. Janssens-Maenhout, M. Osses, S. E. Puliafito, N. Rojas, O. Sánchez-Ccoyllo, S. Tolvett, and R. Y. Ynoue. Evaluation of anthropogenic air pollutant emission inventories for South America at national and city scale. *Atmos. Environ.*, 235:117606, 2020. doi: 10.1016/j.atmosenv.2020.117606.

INE. Censo 2017: Síntesis de resultados. Technical report, Instituto Nacional de Estadísticas, 2018. URL http://www.censo2017.cl/descargas/home/sintesis-de-resultados-censo2017.pdf.

H. Jorquera and F. Barraza. Source apportionment of ambient $PM_{2.5}$ in Santiago, Chile: 1999 and 2004 results. *Sci. Total Environ.*, 435-436:418–429, 2012. doi: 10.1016/j.scitotenv.2012.07.049.

I. G. Kavouras, P. Koutrakis, F. Cereceda-Balic, and P. Oyola. Source Apportionment of PM10 and PM25 in Five Chilean Cities Using Factor Analysis. *J Air Waste Manag Assoc*, 51:451–464, 2001. doi: 10.1080/10473289.2001.10464273.

A. Montecinos and P. Aceituno. Seasonality of the ENSO-Related Rainfall Variability in Central Chile and Associated Circulation Anomalies. *J. Clim.*, 16:281–296, 2003. doi: 10.1175/1520-0442(2003)016¡0281:SOTERR¿2.0.CO;2.

F. Moreno, E. Gramsch, P. Oyola, and M. A. Rubio. Modification in the Soil and Traffic-Related Sources of Particle Matter between 1998 and 2007 in Santiago de Chile. *J Air Waste Manag Assoc*, 60:1410–1421, 2010. doi: 10.3155/1047-3289.60.12.1410.

S. E. Puliafito, D. G. Allende, P. S. Castesana, and M. F. Ruggeri. High-resolution atmospheric emission inventory of the argentine energy sector. Comparison with edgar global emission database. *Heliyon*, 3:e00489, 2017. doi: 10.1016/j.heliyon.2017.e00489.

D. A. Rahn and R. D. Garreaud. A synoptic climatology of the near-surface wind along the west coast of south america. *Int. J. Climatol.*, 34:780–792, 2014. doi: 10.1002/joc.3724.

P. M. Rowe, R. R. Cordero, S. G. Warren, E. Stewart, S. J. Doherty, A. Pankow, M. Schrempf, G. Casassa, J. Carrasco, J. Pizarro, S. MacDonell, A. Damiani, F. Lambert, R. Rondanelli, N. Huneeus, F. Fernandoy, and S. Neshyba. Black carbon and other light-absorbing impurities in snow in the Chilean Andes. *Sci. Rep.*, 9:4008, 2019. doi: 10.1038/s41598-019-39312-0.

R. Urrutia-Jalabert, M. González, A. Gonzalez Reyes, A. Lara, and R. Garreaud. Climate variability and forest fires in central and south-central chile. *Ecosphere*, 9:e02171, 2018. doi: 10.1002/ecs2.2171.